# FedLASE: Performance-Balanced System-Heterogeneous FL via Layer-Adaptive Submodel Extraction

## Abstract

Federated Learning (FL) has gained significant attention for its privacy-preserving capabilities in distributed learning environments. However, the inherent system heterogeneity across edge devices brings significant challenges in deploying a unified global model. Although many submodel extraction methods are designed to address these challenges by selecting a subset of parameters from the global model to accommodate client constraints, our experiments show that existing submodel extraction methods exhibit significant performance discrepancies between submodels with different resource levels, limiting the overall performance of the federated learning system. To overcome these limitations, we propose FedLASE – a novel Layer-Adaptive Submodel Extraction framework that selects important parameters while preserving the structural integrity of the client models, thereby achieving balanced performance across heterogeneous FL clients and improving the convergence. Specifically, our approach quantifies layer importance based on parameter importance and hierarchically extracts critical parameters within each layer while strictly satisfying resource constraints. Theoretically, we rigorously analyze the convergence of FedLASE and investigate the influence of system heterogeneity on its performance. Extensive experiments demonstrate the superiority of FedLASE over the state-of-the-art methods and its robustness across various system-heterogeneous scenarios.

## 1 Introduction

Federated Learning [1, 2] has emerged as a powerful framework for decentralized machine learning, allowing multiple clients, such as mobile devices or Internet of Things systems, to collaboratively train machine learning models without sharing their private data. This approach ensures data privacy and security, as the data remains on the client devices while only model updates are shared. Given the increasing prevalence of edge computing and the growing concerns around data privacy [3, 4], FL has gained significant attention as a practical solution for training large-scale models across a diverse set of clients [5, 6, 7, 8]. However, real-world FL systems are often challenged by system heterogeneity [9, 10, 11], where clients possess different computational resources, storage capacities, and network bandwidth. For simplicity, we characterize the system heterogeneity by the proportion of the model that a client can accommodate relative to the full model, as defined in Definition 1. While high-resource clients can accommodate full-scale deep learning models, resource-constrained clients, such as mobile devices or embedded systems, struggle to train large models effectively. This imbalance leads to inefficient utilization of computational resources and suboptimal model performance.

Submitted to 39th Conference on Neural Information Processing Systems (NeurIPS 2025). Do not distribute.

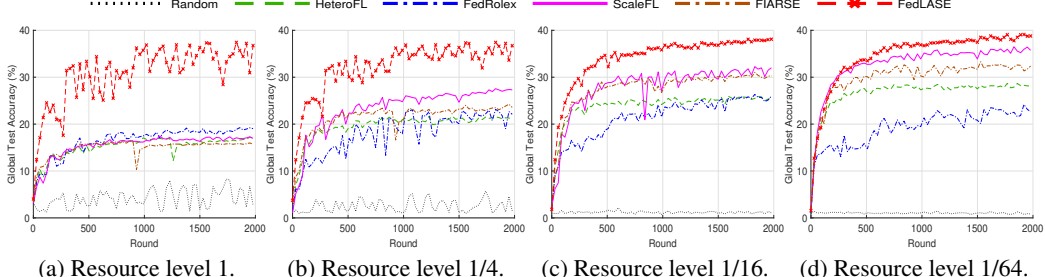

Figure 1: Convergence of different methods across all client resource levels for CIFAR-100 and heterogeneous system $\{1, 1/4, 1/16, 1/64\}\_\{5, 10, 25, 60\}$, showing that the performance gap across resource levels for SOTA methods varies significantly, especially for larger clients with sufficient resources but fewer in number (see (a)), while our method exhibits a more balanced performance.

\* The heterogeneous system $\{1, 1/4, 1/16, 1/64\}\_\{5, 10, 25, 60\}$ has four distinct resource levels: 5 clients capable of running the full model (size 1), 10 clients operating with a reduced model of size 1/4, 25 clients using a smaller model of size 1/16, and 60 clients assigned the smallest model of size 1/64, as shown in Definition 1.

To address system heterogeneity, existing solutions can be broadly categorized into three categories. The first category discards resource-constrained clients or limits the model architecture to the weakest client [12, 13], thereby ensuring system-wide uniformity, but at the cost of underutilizing available computational or data resources. The second category assigns separate models to different client groups based on their computational capacities [14, 15, 16]. Although this enables clients to train models suited to their resources, aggregating models of different sizes and architectures is inherently challenging, especially for knowledge distillation-based approaches, which often require additional public datasets, complicating training and posing privacy risks. The third category, submodel extraction methods [9, 17, 18, 19, 20, 21, 22, 23], provides a more flexible solution by extracting smaller submodels from a shared global model. This method allows clients to participate regardless of resource constraints while maintaining a unified global model.

Among these methods, submodel extraction has gained increasing attention due to its ability to balance model flexibility and consistency. Various extraction techniques have been proposed, ranging from random selection (e.g., Federated Dropout [17]) to static submodel assignment (e.g., HeteroFL [9], FjORD [24]). Although static submodel assignment methods improve training stability compared to random selection, they limit the adaptability of submodels to different clients, often leading to inefficient parameter utilization. FedRolex [18] alleviated this issue by introducing a rolling extraction strategy to improve parameter coverage, while methods such as ScaleFL [25] and DepthFL [13] constructed submodels based on predefined width and depth constraints, incorporating self-distillation to enhance knowledge transfer. However, the aforementioned methods treat all parameters equally, lacking a principled mechanism to determine which parameters should be extracted. Recently, Wu et al. [21] introduced an importance-aware extraction method that ranks parameters globally based on their magnitudes. Nevertheless, this method overlooks inter-layer discrepancies, leading to excessive pruning in certain layers and disrupting the structural integrity of smaller submodels. Our experiment presented in Fig. 1 reveals that existing state-of-the-art (SOTA) submodel extraction methods exhibit significant performance discrepancies across different resource levels, leading to suboptimal performance due to the difficulty of sufficiently utilizing the information of other clients. These findings indicate that treating all layers uniformly or relying solely on a global ranking strategy is insufficient, highlighting the need for a more structured approach that takes into account both layer importance and parameter importance during the submodel extraction process.

Based on these observations, we propose FedLASE (shown in Fig. 2), a novel Layer-Adaptive Submodel Extraction framework designed to balance client performance in system-heterogeneous federated learning by preserving the *structural integrity* of the network architecture through layer-wise extraction of important parameters. Unlike existing methods that rely on global ranking or uniform selection, FedLASE dynamically extracts submodels by incorporating both *layer importance* and *parameter importance*, ensuring that critical structural components are retained across different client resource levels. This leads to more stable training, improved convergence, and enhanced performance, particularly in heterogeneous federated learning environments that more accurately

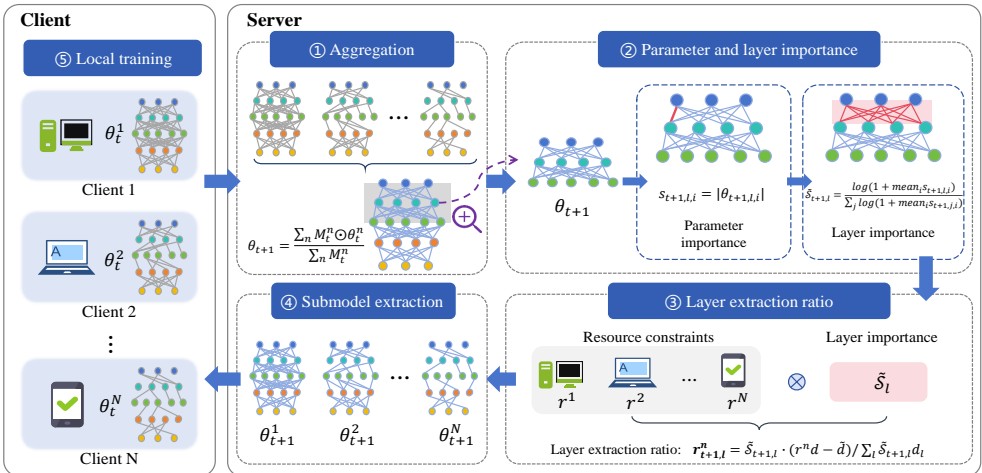

Figure 2: The framework diagram of FedLASE. The server first aggregates the models uploaded by clients to update the global model (①), calculates the importance of each parameter and layer (②), determines the layer extracting ratios based on client resources and layer importance (③), then extracts submodels based on extracting ratios (④) and sends them the clients for local training (⑤).

reflect real-world scenarios, where the number of resource-rich clients is limited and the majority are resource-constrained.

The key contributions of this paper are as follows:

- We propose a novel importance-aware layer-adaptive submodel extraction framework (FedLASE) that enables efficient training across all clients in system-heterogeneous FL.
- We show that adaptively selecting parameters based on layer importance and parameter importance can ensure the preservation of critical structural components across all resource levels, thus balancing the performance of submodels and improving convergence.
- We provide a rigorous proof that FedLASE converges at a rate of $\mathcal{O}(\frac{1}{\sqrt{T}})$, and discuss the impact of system heterogeneity on convergence. To the best of our knowledge, this is the first time to analyze the impact of system heterogeneity on the convergence rate in system-heterogeneous FL.
- Extensive experiments demonstrate the superiority of FedLASE over the existing SOTA methods in terms of both stability and accuracy under various system heterogeneity scenarios, validating its effectiveness in real-world federated learning applications.

The remainder of this paper is organized as follows. Section 2 introduces the standard formulation of FL and extends it to the system-heterogeneous setting. Section 3 provides a detailed description of the proposed FedLASE framework. Theoretical analysis is presented in Section 4, while Section 5 reports extensive experimental results that demonstrate the effectiveness and superiority of FedLASE. Finally, Section 6 concludes this paper and outlines potential directions for future research.

## 2 Preliminaries

In this section, we first introduce the standard formulation of FL and then extend it to the system-heterogeneous setting, which serves as the foundation of our method in subsequent sections.

The objective of traditional FL is to optimize a global model $\theta \in \mathbb{R}^d$ by minimizing the aggregated loss across $N$ clients [1, 7], i.e.,

$$\min_{\theta} F(\theta) \triangleq \sum_{n=1}^{N} p_n F_n(\theta),$$

where $F_n(\theta) = \sum_{k=1}^{m_n} l(\theta; d_k^n)/m_n$ represents the local objective function for client $n$, $l(\cdot)$ is the loss function, $p_n$ denotes the aggregation weight, the term $d_k^n$ corresponds to the $k$th data sample

of client $n$, and $m_n$ is the total number of local training samples for client $n$. To accommodate the diverse computational capabilities of clients in real-world FL scenarios, system-heterogeneous FL allows each client to train a submodel suited to its resource constraints. To formalize this extension and analyze the impact of system heterogeneity (shown in Section 4), we first give a definition of heterogeneous system in federated learning.

*Definition 1:* (Heterogeneous System) In federated learning setting, a heterogeneous system denoted by $\{\text{level}_1, \text{level}_2, \ldots, \text{level}_p\}_-\{N_1, N_2, \ldots, N_p\}$ consists of $p$ resource levels $\{\text{level}_1, \text{level}_2, \ldots, \text{level}_p\}$ and the $i$th resource level is allocated $N_i$ clients with $\text{level}_i \in (0, 1]$ representing the fraction of the global model that clients at this level can accommodate and $\sum_{i=1}^p N_i = N$.

Based on this definition, we now turn to system-heterogeneous federated learning. Denote the resource capacity of client $n$ by $r^n \in \{\text{level}_1, \text{level}_2, \ldots, \text{level}_p\}$. Then the submodel for client $n$ can be constructed by applying a binary mask $M^n \in \{0, 1\}^d$ to the global model $\theta$

$$\theta^n = \theta \odot M^n,$$

where $\odot$ represents element-wise multiplication, $M_i^n = 1$ means that the $i$th parameter is retained, and $M_i^n = 0$ means that it is pruned. Obviously, the number of retained parameters in each submodel satisfies $\|\theta^n\|_0 \leq r^n d$. Under this system-heterogeneous FL setting, the global objective can be reformulated as:

$$\min_{\theta, M^1, M^2, \ldots, M^N} \sum_{n=1}^N p_n \tilde{F}_n(\theta \odot M^n) = \sum_{n=1}^N p_n \tilde{F}_n(\theta^n),$$

where $\tilde{F}_n(\theta^n) = \sum_{k=1}^{m_n} l_n(\theta^n; d_k^n)/m_n$. For simplicity, we assume that all clients are equally weighted in the aggregation process, i.e., $p_n = 1/N$.

# 3 FedLASE: Importance-aware Layer-adaptive Submodel Extraction

In system-heterogeneous federated learning, extracting an appropriate submodel for each client is crucial for balancing computational resources with model expressiveness. However, existing submodel extraction methods often overlook the differences of parameters in different layers, resulting in the loss of critical information and reduced representational capacity of the submodels.

To address these limitations, we propose FedLASE, an importance-aware layer-adaptive submodel extraction framework that dynamically extracts parameters at each layer based on parameter importance and layer importance. The overall framework is presented in Fig. 2, and the corresponding algorithm is provided in Algorithm 1 (shown in the Appendix B due to space limitations). Specifically, to achieve effective submodel extraction while maintaining model integrity, FedLASE first evaluates the importance of each parameter and layer in the aggregated global model, identifying the most critical components for extraction (shown in Fig. 2 (②)). Then, leveraging the computed importance scores along with client resource constraints, the server determines appropriate layer-wise extraction ratios for each client (shown in Fig. 2 (③)), ensuring that submodels remain computationally feasible while preserving the essential structural information of the network architecture. Based on these extraction ratios, important parameters are selectively extracted from each layer to form client-specific submodels (shown in Fig. 2 (④)), which are subsequently trained locally and aggregated (shown in Fig. 2 (⑤) and (①)) to refine the global model. In the following subsections, we provide a detailed explanation of each component.

## 3.1 Importance Measurement for Parameters and Layers

Existing research indicates that the magnitude of model parameters can serve as an effective indicator of their importance [26, 27], with parameters having higher absolute values generally exhibiting a greater impact on the expressiveness of the model. Although there are alternative metrics for the estimation of parameter importance [28, 29, 30, 31], we adopt the magnitude-based criterion for its simplicity and computational efficiency.

Unlike previous methods that rank all parameters globally, FedLASE calculates importance scores within each layer to preserve structural integrity and avoid excessive pruning in certain layers. Specifically, for the $i$th parameter $\theta_{l,i}$ in the $l$th layer of the global model $\theta$, its importance score is measured by $s_{l,i} = |\theta_{l,i}|$. In this paper, we measure the importance of the $l$th layer (denoted by $\mathcal{S}_l$)

using the mean importance score of the parameters within that layer, i.e., $\mathcal{S}_l = \text{mean}_i s_{l,i}$. To mitigate dominance by extreme values while maintaining relative importance relationships, we normalize layer importance using the following logarithmic transformation to ensure a more balanced allocation of the extracted parameters between layers:

$$\tilde{\mathcal{S}}_l = \frac{\log(1 + \mathcal{S}_l)}{\sum_j \log(1 + \mathcal{S}_j)}.$$

## 3.2 Layer-adaptive Submodel Extraction

After obtaining the layer importance scores, another crucial aspect is determining the extraction ratio for each layer across different clients, ensuring that the resource constraints of each client are satisfied. Let $r^n$ denote the fraction of the global model allocated to client $n$, implying that the number of parameters extracted by client $n$ from the global model will not exceed $d^n \triangleq r^n d$ with $d$ being the total number of parameters in the global model. Considering the fact that the *first layer*, *last layer*, *normalization layers*, and *bias terms* are crucial for preserving input representations, stabilizing training, and maintaining expressiveness, especially in smaller submodels [32, 31], we fully retain these components. Let $\tilde{d}$ represent the number of parameters retained due to these prior constraints. The remaining parameters available for extraction are then bounded by $d^n - \tilde{d}$, with the assumption that $d^n > \tilde{d}$.

Denote the set of prunable layers as $\{l_1, l_2, \ldots, l_L\}$. To allocate extraction ratios according to the importance of each layer, we assign a higher extraction ratio to more critical layers. Therefore, based on the resource limitation of clients, we assume the extraction ratio of the $l_i$th layer for client $n$ as

$$r_{l_i}^n = \alpha^n \tilde{\mathcal{S}}_{l_i}, \quad (\alpha^n \geq 0).$$

To ensure the submodel satisfy the resource budget of client $n$, the following inequality should be satisfied:

$$r_{l_1}^n d_{l_1} + r_{l_2}^n d_{l_2} + \cdots + r_{l_L}^n d_{l_L} \leq d^n - \tilde{d},$$

where $d_{l_i}$ is the number of parameters in the $l_i$th layer of the global model $\theta$, excluding biases. Thus, the upper bound of $\alpha^n$ is

$$\alpha^n \leq (d^n - \tilde{d}) / \Big(\sum_{i=1}^{L} \tilde{\mathcal{S}}_{l_i} d_{l_i}\Big).$$

For simplicity, we can set the importance-aware extraction ratio of each layer for client $n$ as

$$r_{l_i}^n = \tilde{\mathcal{S}}_{l_i} \cdot (d^n - \tilde{d}) / \Big(\sum_{i=1}^{L} \tilde{\mathcal{S}}_{l_i} d_{l_i}\Big). \tag{1}$$

After getting the layer-wise extraction ratios for each client, we extract the top $r_{l_i}^n \cdot d_{l_i}$ parameters in the $l_i$th layer based on their importance. This results in a threshold value $\tilde{\theta}_{l_i}^n$ and a corresponding mask $M_{l_i}^n$ for the $l_i$th layer, which together define the extracted submodel for client $n$.

By incorporating prior constraints on key structural components and adapting extraction ratios based on layer importance, our method ensures the retention of essential information for each client, balancing the performance across submodels and enhancing both convergence and robustness in various heterogeneous FL environments.

## 3.3 Local Training Optimization and Submodel Aggregation

To refine submodel extraction and improve the efficiency of local training, we integrate the straight-through estimation (STE) technique [21, 33, 34] into the local training process. This method enhances gradient flow by sharpening the distinction between important and less important parameters. Specifically, to obtain the submodel for client $n$, we use the probability $\text{clip}\big((\theta_{l,j} - \tilde{\theta}_l^n)/(\theta_{l,j} + \tilde{\theta}_l^n), 0, 1\big)$ to set the mask for the $j$th parameter in the $l$th layer $\theta_{l,j}$ to 1 with $\tilde{\theta}_l^n$ being the extraction threshold in the $l$th layer for client $n$. Then, the $l$th layer of the gradient updated during the local training process for client $n$ is adjusted as

$$\Big(\nabla_\theta F_n(\theta \odot M^n)\Big)_l = \Big(\nabla F_n(\theta \odot M^n)\Big)_l \odot M_l^n \odot \left(1 + \frac{2|\theta_l|\tilde{\theta}_l^n}{(|\theta_l| + \tilde{\theta}_l^n)^2}\right). \tag{2}$$

The derivation of Eq. (2) is similar to that of Eq. (3) in [21], and thus is omitted.

After local training, each client uploads its trained submodel to the server. Due to the model heterogeneity introduced by the layer-wise extraction process, different clients retain different subsets of model parameters. To protect the personalization of the subnetworks, we adopt the following overlapping averaging strategy [27, 35]

$$\theta = \Big( \sum_n M^n \odot \theta^n \Big) \Big/ \Big( \sum_n M^n \Big).$$  (3)

This strategy ensures that each parameter in the global model is updated based only on clients that have retained and trained it, preventing issues arising from missing updates in pruned parameters and preserving the personalization of clients.

## 3.4 Complexity Analysis

In the final of this section, we conduct a comparative analysis of computational and communication efficiency between FedLASE and the SOTA methods, demonstrating that FedLASE achieves a balanced computational and communication complexity compared with the SOTA methods. Detailed discussion is presented in Appendix C due to space limitations.

# 4 Theoretical Analysis

To theoretically evaluate the impact of system heterogeneity, we introduce a new assumption about model noise reduction based on Definition 1. This assumption extends the concept in [19], aiming to quantify the noise introduced by each client due to the submodel extraction process, which is related to its resource levels.

*Assumption 1:* (Model Reduction Noise) For heterogeneous system $\{\text{level}_1, \text{level}_2, \dots, \text{level}_p\}_-$ $\{N_1, N_2, \dots, N_p\}$, assume that there exist some constants $\delta_i \geq 0$ such the model reduction noise for the client with $\text{level}_i$ is bounded by

$$\|\theta_t - \theta_t \cdot M_t^{\text{level}_i}\|^2 \leq (1 - \text{level}_i)\delta_i^2\|\theta_t\|^2,$$  (4)

where $M_t^{\text{level}_i}$ is the mask for the $i$th resource level in round $t$.

Obviously, a higher resource level means less model reduction noise. When the mask is generated by globally sorting the parameters based on their magnitudes and $\text{level}_i \cdot d$ is an integer, it is easy to prove that equality holds in Eq. (4) for $\delta_i = 1$. Thus, the above assumption is well-defined.

Based on Assumption 1 and the standard Assumptions 2-5 outlined in Appendix D.2, we establish the following convergence theorem, and its proof is presented in Appendix D.2 for brevity.

*Theorem 1:* Suppose Assumptions 1, 2, 3, 4 or 1, 2, 3, 5 hold and the local learning rate satisfies $\eta = \mathcal{O}(1/(K\sqrt{T}))$ with $K$ and $T$ being the number of local epoch and total round. Then the proposed FedLASE converges to a small neighborhood of a stationary point of the standard FL under heterogeneous system $\{\text{level}_1, \text{level}_2, \dots, \text{level}_p\}_-\{N_1, N_2, \dots, N_p\}$:

$$\frac{1}{T}\sum_{t=0}^{T-1}\sum_{i \in \mathcal{I}_t}\mathbb{E}\Big(\nabla\tilde{F}(\theta_t)\Big)_i^2 \leq \mathcal{O}\Big(\frac{1}{\sqrt{T}}\Big) + \mathcal{O}\Big(\frac{1}{\sqrt{T}}\Big)\frac{1}{T}\sum_{t=0}^{T-1}\sum_{i=1}^{p}N_i(1-\text{level}_i)\delta_i^2\|\theta_t\|^2$$

$$+ \mathcal{O}\Big(\frac{1}{T}\Big)\sum_{t=0}^{T-1}\sum_{i=1}^{p}N_i(1-\text{level}_i)\delta_i^2\|\theta_t\|^2,$$  (5)

where $\mathcal{I}_t$ is the index set of elements updated in the $t$th round.

*Remark 1:* For the ideal environment in which all clients have sufficient resources to train the full model, i.e., the system $\{1\}_-\{N\}$, the last two terms on the right-hand side of Eq. (5) become zero. Therefore, FedLASE converges with a rate of $\mathcal{O}(1/\sqrt{T})$. In heterogeneous client resource settings, since $\frac{1}{T}\sum_{t=0}^{T-1}\sum_{i=1}^{p}N_i(1-\text{level}_i)\delta_i^2\|\theta_t\|^2$ in Eq. (5) is often bounded [19], FedLASE will converge to a small neighborhood of a stationary point of the standard FL. Moreover, for the fixed client allocation $\{N_1, N_2, \dots, N_p\}$, the convergence upper bound becomes smaller as the resource level increases. In contrast, when the resource level $\{\text{level}_1, \text{level}_2, \dots, \text{level}_p\}$ is fixed, the larger the number of clients with a higher resource level, the smaller the convergence upper bound, as verified in Section 5.

## 5 Experiments

In this section, we evaluate the effectiveness and superiority of our proposed method in system-heterogeneous federated learning. The basic experimental configurations are as follows.

**Datasets and models.** To evaluate the effectiveness of FedLASE, we conduct experiments on two classical image classification datasets: CIFAR-10 and CIFAR-100 [36]. We employ ResNet-18 as the backbone model, replacing batch normalization (BN) layers with static BN [21, 37].

**Data heterogeneity.** To evaluate the impact of data heterogeneity on federated learning, we consider two sets of data distributions across clients: IID distribution and Dirichlet distribution with concentration parameter $\alpha$ (denoted as $\text{Dir}(\alpha)$) [21].

**System heterogeneity.** To evaluate the impact of system heterogeneity, the two sets of *client resource levels* $\{1, 1/4, 1/16, 1/64\}$ and $\{1, 16/25, 9/25, 4/25, 1/25\}$ are considered. Unlike previous studies assuming an equal distribution of clients across all resource levels, we explore multiple allocation strategies to better reflect real-world scenarios, where resource-rich clients are relatively scarce while resource-constrained clients are more prevalent. Specifically, we consider three different *client allocation schemes* for 100 clients: $\{5, 10, 25, 60\}$, $\{10, 20, 30, 40\}$, and $\{25, 25, 25, 25\}$ for the four-level setting, as well as $\{5, 5, 10, 20, 60\}$, $\{5, 10, 15, 20, 50\}$, and $\{20, 20, 20, 20, 20\}$ for the five-level setting.

**Baselines.** To evaluate the effectiveness of our approach, we compare it with the SOTA submodel extraction methods: HeteroFL [9], FedRolex [18], ScaleFL [25], FIARSE [21] and a simple random baseline where the parameters are extracted randomly in each layer with equal proportion, while fully preserving the first and last layers.

**Experimental setup.** To ensure a fair and comprehensive evaluation, we adopt the standardized training procedure across all methods. In each communication round, $10\%$ of the 100 clients are randomly selected to participate in training. The training process spans 2000 communication rounds, with each selected client performing 5 local epochs per round using a batch size of 20, as specified in [21]. The default data partitioning follows a Dirichlet distribution with $\alpha = 0.1$. For optimization, we employ SGD with momentum. The learning rate is selected from $\{0.01, 0.1\}$, while the momentum coefficient is chosen from $\{0.0, 0.8, 0.9\}$. All experiments were conducted on 2 NVIDIA GeForce RTX 4090 GPUs.

**Evaluation.** For performance evaluation, we aggregate the test datasets of all clients to form a global test set. By default, all results correspond to the best-performing hyperparameter configuration. To ensure robustness and stability, we report the average Top-1 accuracy over the last 20 communication rounds, mitigating potential performance fluctuations. Each experiment is repeated three times with different random seeds, and the final results are presented as the average accuracy across these runs.

### 5.1 Performance Comparison with Baselines

**Local Test Accuracy (AccL) and Global Test Accuracy (AccG).** To evaluate the effectiveness and generalization of FedLASE, we compare its performance against the state-of-the-art methods from two perspectives: local test accuracy and global test accuracy. The results of AccL and AccG for different methods under all client resource levels are summarized in Table 1. For AccL, FedLASE consistently outperforms existing methods across all system heterogeneity settings, achieving the highest average accuracy in all resource levels. Specifically, FedLASE achieves an average AccL of $41.95\%$ and $41.35\%$ under two sets of system-heterogeneous scenarios, which are higher than the second-best method by $5.55\%$ and $4.82\%$, respectively. Notably, for the highest resource level (i.e., resource level 1), our method outperforms the second-best method by $22.65\%$ and $14.94\%$, demonstrating the effectiveness of FedLASE to enhance the model performance of resource-rich clients, even when the number of such clients is limited. For AccG, FedLASE still maintains clear superiority, surpassing the second-best method by $9.56\%$ and $7.88\%$ in average global accuracy under two sets of scenarios. These results further highlight its superiority and strong generalization capability.

**Balanced Client Performance.** To provide a more intuitive understanding of the advantages for our method, we analyze the convergence behavior of different approaches across all client resource levels for heterogeneous system $\{1, 1/4, 1/16, 1/64\}\_\{5, 10, 25, 60\}$, as shown in Fig. 1. From these

Table 1: Comparison of accuracy for different methods across all client resource levels under CIFAR-100 and two sets of heterogeneous systems.

| Scenario | Method | Mean | | Resource level 1 | | Resource level 1/4 | | Resource level 1/16 | | Resource level 1/64 | |
|---|---|---|---|---|---|---|---|---|---|---|---|
| | | AccL | AccG | AccL | AccG | AccL | AccG | AccL | AccG | AccL | AccG |
| {1, 1/4, 1/16, 1/64} _ {5, 10, 25, 60} | Random | 1.20 | 1.91 | 2.82 | 3.56 | 1.64 | 1.92 | 1.01 | 1.14 | 1.07 | 1.03 |
| | HeteroFL | 28.01 | 22.98 | 14.71 | 16.99 | 24.01 | 21.21 | 26.28 | 25.41 | 30.51 | 28.31 |
| | FedRolex | 24.70 | 22.61 | 18.27 | 19.31 | 24.66 | 22.92 | 26.16 | 24.82 | 24.63 | 23.38 |
| | ScaleFL | 36.40 | 27.78 | 14.37 | 17.18 | 30.60 | 27.14 | 34.26 | 31.23 | 40.10 | 35.56 |
| | FIARSE | 32.45 | 25.35 | 11.91 | 15.80 | 24.92 | 23.71 | 31.92 | 29.82 | 35.47 | 32.06 |
| | FedLASE | 41.95 | 37.36 | 40.92 | 36.26 | 41.56 | 36.94 | 40.90 | 37.78 | 42.55 | 38.65 |
| {1, 1/4, 1/16, 1/64} _ {10, 20, 30, 40} | Random | 2.23 | 3.36 | 6.54 | 7.17 | 3.78 | 3.89 | 1.29 | 1.29 | 1.08 | 1.10 |
| | HeteroFL | 27.48 | 25.61 | 19.77 | 21.22 | 29.10 | 27.14 | 28.33 | 28.47 | 27.97 | 25.64 |
| | FedRolex | 26.54 | 25.26 | 25.74 | 25.31 | 30.38 | 27.93 | 30.01 | 27.66 | 22.21 | 20.15 |
| | ScaleFL | 36.53 | 31.31 | 22.93 | 23.37 | 35.78 | 32.03 | 36.10 | 34.34 | 40.63 | 35.49 |
| | FIARSE | 33.42 | 29.69 | 21.92 | 22.90 | 32.69 | 30.11 | 35.22 | 34.30 | 35.31 | 31.47 |
| | FedLASE | 41.35 | 39.19 | 40.68 | 38.33 | 41.22 | 39.10 | 42.17 | 41.40 | 40.96 | 37.93 |

\* Mean: the average accuracy of all resource levels; AccL/AccG: the local/global test accuracy.

\* The methods marked in **bold** and underlined represent the best-performing methods and second-best methods, respectively.

Table 2: Comparison of average AccG for different methods across two sets of heterogeneous systems.

| Dataset | Method | {1, 1/4, 1/16, 1/64} | | | {1, 16/25, 9/25, 4/25, 1/25} | | |
|---|---|---|---|---|---|---|---|
| | | {5, 10, 25, 60} | {10, 20, 30, 40} | {25, 25, 25, 25} | {5, 5, 10, 20, 60} | {5, 10, 15, 20, 50} | {20, 20, 20, 20, 20} |
| CIFAR-10 | Random | 10.15 (↓ 66.38) | 10.79 (↓ 68.55) | 20.40 (↓ 59.27) | 11.68 (↓ 71.19) | 13.28 (↓ 69.52) | 45.94 (↓ 36.55) |
| | HeteroFL | 61.41 (↓ 15.12) | 65.64 (↓ 13.70) | 73.88 (↓ 5.79) | 65.63 (↓ 17.24) | 67.87 (↓ 14.93) | 72.51 (↓ 9.98) |
| | FedRolex | 58.31 (↓ 18.22) | 65.42 (↓ 13.92) | 65.09 (↓ 14.58) | 69.71 (↓ 13.16) | 71.82 (↓ 10.98) | 73.75 (↓ 8.74) |
| | ScaleFL | 52.52 (↓ 24.01) | 57.28 (↓ 22.06) | 61.72 (↓ 17.95) | 50.97 (↓ 31.90) | 55.89 (↓ 26.91) | 61.07 (↓ 21.42) |
| | FIARSE | 61.60 (↓ 14.93) | 72.04 (↓ 7.30) | 79.05 (↓ 0.62) | 72.37 (↓ 10.50) | 75.65 (↓ 7.15) | 79.59 (↓ 2.90) |
| | FedLASE | 76.53 | 79.34 | 79.67 | 82.87 | 82.80 | 82.49 |
| CIFAR-100 | Random | 1.91 (↓ 35.44) | 3.36 (↓ 35.81) | 12.67 (↓ 26.12) | 5.01 (↓ 41.14) | 10.06 (↓ 34.94) | 25.13 (↓ 19.93) |
| | HeteroFL | 22.98 (↓ 14.37) | 25.61 (↓ 13.56) | 27.87 (↓ 10.92) | 23.94 (↓ 22.21) | 25.34 (↓ 19.66) | 26.67 (↓ 18.39) |
| | FedRolex | 22.61 (↓ 14.74) | 25.26 (↓ 13.91) | 28.12 (↓ 10.67) | 29.03 (↓ 17.12) | 30.17 (↓ 14.83) | 33.39 (↓ 11.67) |
| | ScaleFL | 27.78 (↓ 9.57) | 31.31 (↓ 7.86) | 35.29 (↓ 3.50) | 27.03 (↓ 19.12) | 29.46 (↓ 15.54) | 34.93 (↓ 10.13) |
| | FIARSE | 25.32 (↓ 12.03) | 29.69 (↓ 9.48) | 35.10 (↓ 3.69) | 30.15 (↓ 16.00) | 33.38 (↓ 11.62) | 38.13 (↓ 6.93) |
| | FedLASE | 37.35 | 39.17 | 38.79 | 46.15 | 45.00 | 45.06 |

\* Resource levels: {1, 1/4, 1/16, 1/64} and {1, 16/25, 9/25, 4/25, 1/25}.

\* Client allocation schemes: {5, 10, 25, 60}, {10, 20, 30, 40}, {25, 25, 25, 25}, {5, 5, 10, 20, 60}, {5, 10, 15, 20, 50}, {20, 20, 20, 20, 20}.

\* The values in parentheses indicate the accuracy reduction relative to our method.

results, we observe that FedLASE exhibits more stable performance across different resource levels, whereas existing methods suffer from significant performance gaps between high and low resource levels. Another key observation is that larger submodels in SOTA methods tend to underperform compared to smaller ones, despite being deployed on resource-rich clients. This counterintuitive behavior results from an imbalance in training updates: smaller submodels, hosted on the majority of resource-constrained clients, receive more frequent updates, while larger submodels, trained on fewer high-resource clients, are updated less frequently, leading to suboptimal learning. By contrast, FedLASE mitigates this issue through its importance-aware layer-adaptive submodel extraction strategy. By prioritizing essential parameters at each layer, FedLASE ensures that all submodels retain critical structural information, allowing large submodels to maintain competitive performance without compromising small submodel efficiency.

## 5.2 Impact of System Heterogeneity

To systematically investigate the impact of system heterogeneity, we extend our evaluation beyond CIFAR-100 to additional datasets and heterogeneous systems, as detailed in Table 2. In most system settings, the random method fails to converge, highlighting the inherent difficulty of achieving stable learning in highly imbalanced environments. This challenge becomes even more pronounced in realistic federated learning scenarios, where high-performance clients are scarce and the majority of participating clients possess only limited computational resources.

From Table 2, it can be seen that FedLASE consistently outperforms SOTA methods, achieving significantly superior test accuracy. This demonstrates the robustness of our approach in various system-heterogeneous federated learning environments. Moreover, more clients with high resource levels often result in better performance, verifying the statement in Remark 1. Notably, one can find that existing SOTA methods exhibit substantial performance fluctuations as the proportion of

resource-constrained clients increases. For example, the test accuracy of FIARSE for CIFAR-100 drops sharply from 38.13% to 30.15% in the second set of resource level settings, demonstrating the instability caused by inefficient adaptation to clients with vastly different computational capabilities. In contrast, FedLASE maintains significantly more stable accuracy, with fluctuations constrained between 45% and 46.15% across different client distributions. This stability is attributed to our layer-wise adaptive parameter extraction, which ensures submodels consistently retain critical structural components. By prioritizing key parameters within each layer, FedLASE prevents excessive pruning in essential layers, thereby mitigating the adverse effects of system heterogeneity.

## 5.3 Impact of Data Heterogeneity

To examine the effect of data heterogeneity, we perform comparative experiments with the SOTA methods under different data heterogeneity settings, including IID and Dirichlet distributions, as shown in Table 3. It can be seen that as the degree of data heterogeneity increases, the performance of all methods decreases. Notably, FedLASE consistently achieves higher accuracy than the recent methods FIARSE and ScaleFL in all settings, with a particularly significant improvement in highly non-IID scenarios. These results illustrate that our importance-aware layer-adaptive extraction strategy can enhance model robustness under diverse data distributions.

Table 3: Comparison of global test accuracy for different methods across various data distributions under heterogeneous system $\{1, 1/4, 1/16, 1/64\}$ _ $\{10, 20, 30, 40\}$.

| Method | CIFAR-10 | | | CIFAR-100 | | |
|---|---|---|---|---|---|---|
| | iid | Dir(0.3) | Dir(0.1) | iid | Dir(0.3) | Dir(0.1) |
| HeteroFL | 77.68 ($\downarrow$ 6.53) | 72.11 ($\downarrow$ 6.93) | 65.64 ($\downarrow$ 13.70) | 31.25 ($\downarrow$ 13.81) | 29.45 ($\downarrow$ 12.75) | 25.61 ($\downarrow$ 13.56) |
| FedRolex | 77.49 ($\downarrow$ 6.72) | 68.26 ($\downarrow$ 10.78) | 65.42 ($\downarrow$ 13.92) | 35.05 ($\downarrow$ 10.01) | 31.26 ($\downarrow$ 10.94) | 25.26 ($\downarrow$ 13.91) |
| ScaleFL | 80.87 ($\downarrow$ 3.34) | 68.60 ($\downarrow$ 10.44) | 57.28 ($\downarrow$ 22.06) | 42.62 ($\downarrow$ 2.44) | 38.01 ($\downarrow$ 4.19) | 31.31 ($\downarrow$ 7.86) |
| FIARSE | 82.64 ($\downarrow$ 1.57) | 77.75 ($\downarrow$ 1.29) | 72.04 ($\downarrow$ 7.30) | 37.03 ($\downarrow$ 8.03) | 34.04 ($\downarrow$ 8.16) | 29.69 ($\downarrow$ 9.48) |
| FedLASE | **84.21** | **79.04** | **79.34** | **45.06** | **42.20** | **39.17** |

## 5.4 Impact of Network Architecture

This subsection further investigates the impact of different network architectures. The experimental results shown in Table 4 demonstrate that our method consistently outperforms existing methods across two distinct network architectures under two sets of heterogeneous scenarios. In particular, for the heterogeneous system $\{1, 1/4, 1/16, 1/64\}$ _ $\{5, 10, 25, 60\}$, our method achieves a notable performance improvement on CIFAR-100, surpassing the second-best method by 9.57% and 11.79% for the two network architectures, respectively. This further validates the robustness of our approach, highlighting its adaptability to different network architectures in real-world scenarios.

Table 4: Comparison of global test accuracy for different methods on CIFAR-10 and CIFAR-100 using ResNet-18 and ResNet-34, under two heterogeneous system settings.

| Method | $\{1, 1/4, 1/16, 1/64\}$ _ $\{5, 10, 25, 60\}$ | | | | $\{1, 1/4, 1/16, 1/64\}$ _ $\{10, 20, 30, 40\}$ | | | |
|---|---|---|---|---|---|---|---|---|
| | CIFAR-10 | | CIFAR-100 | | CIFAR-10 | | CIFAR-100 | |
| | ResNet-18 | ResNet-34 | ResNet-18 | ResNet-34 | ResNet-18 | ResNet-34 | ResNet-18 | ResNet-34 |
| ScaleFL | 52.52 ($\downarrow$ 24.01) | 46.45 ($\downarrow$ 24.10) | 27.78 ($\downarrow$ 9.57) | 25.72 ($\downarrow$ 12.41) | 57.28 ($\downarrow$ 22.06) | 51.65 ($\downarrow$ 23.97) | 31.32 ($\downarrow$ 7.85) | 29.84 ($\downarrow$ 9.54) |
| FIARSE | 61.60 ($\downarrow$ 14.93) | 62.63 ($\downarrow$ 7.92) | 25.36 ($\downarrow$ 11.99) | 26.34 ($\downarrow$ 11.79) | 72.04 ($\downarrow$ 7.30) | 66.11 ($\downarrow$ 9.51) | 29.71 ($\downarrow$ 9.46) | 31.95 ($\downarrow$ 7.43) |
| FedLASE | **76.53** | **70.55** | **37.35** | **38.13** | **79.34** | **75.62** | **39.17** | **39.38** |

## 6 Conclusion

In this paper, we proposed the FedLASE framework, an importance-aware layer-adaptive submodel extraction method designed to address the challenges posed by system heterogeneity in federated learning. By considering both parameter importance and layer importance, our method ensures that the critical components in each layer of the global model are preserved, even in resource-constrained environments. Through extensive experiments across different datasets and system-heterogeneous scenarios, we demonstrate that FedLASE significantly outperforms state-of-the-art methods in both global and local test accuracy. In particular, it excels in maintaining stable performance across a wide range of client capacities, ensuring efficient and effective training in heterogeneous FL environments. This illustrates its effectiveness in real-world federated learning scenarios, where clients have different resource capacities. In the future, we will focus on exploring more efficient resource allocation strategies and aggregation schemes to further optimize the performance of system-heterogeneous federated learning, leveraging the characteristics of system heterogeneity.

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

# A Related Work

We systematically review existing approaches to address system heterogeneity in federated learning, categorizing them into three primary strategies: a) client exclusion or model architecture restriction, b) client-specific model training, and c) submodel extraction methods. Our analysis focuses on submodel extraction due to its superior adaptability in heterogeneous environments.

## A.1 Client Exclusion or Model Architecture Restriction

The simplest strategy involves excluding resource-constrained clients or constraining the global model architecture to match the weakest devices [12, 13]. While this approach ensures uniform model architecture across clients and simplifies aggregation, it introduces two critical limitations. First, client exclusion reduces data diversity, potentially inducing model bias and compromising generalization capabilities. Second, architectural constraints prevent high-resource clients from leveraging more complex models that could enhance learning outcomes. These limitations ultimately undermine the system's capacity to utilize available computational resources effectively.

## A.2 Client-Specific Model Training

Alternative approaches enable clients to train models commensurate with their computational capabilities [14, 38, 16, 39]. In this paradigm, high-capacity clients train larger models while resource-limited clients operate smaller variants. However, aggregating heterogeneous model architectures poses significant technical challenges. Knowledge distillation has emerged as a primary solution, where larger teacher models transfer knowledge to smaller student models [40]. Notable implementations include FedDF [14], which distilled knowledge from multiple client classifiers using an additional public dataset, and FedGKT [38], employing group knowledge transfer to enable clients to train small models while a larger model is maintained on the server. Nevertheless, these knowledge distillation-based approaches often depend on additional datasets that may be unavailable due to privacy constraints or domain incompatibility.

## A.3 Submodel Extraction Methods

Unlike the aforementioned approaches, which limit the flexibility of the model or require complex aggregation schemes, the submodel extraction methods allow clients to train smaller models derived from a global model while maintaining a unified architecture between clients. This approach balances adaptability and implementation simplicity, making it particularly suitable for heterogeneous FL systems. For example, inspired by dropout techniques in centralized learning [41], Federated Dropout [17] randomly selected a subset of neurons per layer to form client-specific submodels. Although simple to implement, its randomness leads to unstable training and performance degradation, especially in the case of high system and data heterogeneity, as shown in our experiments. To improve stability, structured submodel extraction methods such as HeteroFL [9] and FjORD [24] predefined fixed submodel assignments for each client. Although this reduces randomness, it restricts data utilization, as different submodels are trained only on specific client subsets, limiting the generalization of the global model. FedRolex [18] alleviated this issue by introducing a rolling submodel extraction strategy, allowing different model segments to be trained over time, thus improving parameter coverage and mitigating model drift. ScaleFL [25] and DepthFL [13] further refined submodel selection based on depth and width configurations, using self-distillation to enhance knowledge transfer between different resource levels. Despite these advancements, most existing methods lack a principled mechanism for parameter selection, treating all model components equally. This often results in suboptimal submodel configurations that fail to retain the most crucial information. To address this, FIARSE [21] introduced an importance-aware approach that globally ranks parameters by importance before extraction. While this strategy demonstrates superior performance compared to uniform selection, it does not consider the variations of parameter importance across different layers. Consequently, certain layers may be excessively pruned in smaller submodels, leading to structural imbalances that degrade model stability and overall performance.

# B  Algorithm of FedLASE

---

**Algorithm 1 FedLASE: Importance-aware Layer-Adaptive Submodel Extraction**

---

**Input:** Local learning rate $\eta$, total round $T$, local epoch $K$, initial global model $\theta_0$, client resource constraints $\{r^n\}$.

1: **for** $t = 0, 1, \ldots, T - 1$ **do**
2:     Sample a set of clients $\mathcal{A} \subseteq [N]$
3:     **Server-side Submodel Extraction:**
4:     **for** each client $n \in \mathcal{A}$ in parallel **do**
5:         Compute layer-wise extraction ratio $r^n_{t,l_i}$ using Eq. (1)
6:         Extract top $r^n_{t,l_i} \cdot d_{l_i}$ most important parameters per layer to obtain mask $M^n_t$ and threshold $\tilde{\theta}^n_{t,l_i}$
7:         Send $\theta_t \odot M^n_t$ and $\tilde{\theta}^n_{t,l_i}$ to client $n$
8:     **end for**
9:     **Client-side Local Training:**
10:     **for** each client $n \in \mathcal{A}$ in parallel **do**
11:         Initialize $\theta^n_{t,0} = \theta_t \odot M^n_t$
12:         **for** $k = 0, \ldots, K - 1$ **do**
13:             Compute gradient using Eq. (2): $g^n_{t,k} = \nabla_{\theta^n_{t,k}} \tilde{F}_n(\theta^n_{t,k} \odot M^n_t)$
14:             Update local model: $\theta^n_{t,k+1} = \theta^n_{t,k} - \eta \cdot g^n_{t,k}$
15:         **end for**
16:         Upload the trained submodel $\theta^n_t \triangleq \theta^n_{t,K}$ to the server
17:     **end for**
18:     **Server-side Model Aggregation:**
19:     Aggregate local models using Eq. (3): $\theta_{t+1} = \left( \sum_{n \in \mathcal{A}} M^n_t \odot \theta^n_t \right) \Big/ \left( \sum_{n \in \mathcal{A}} M^n_t \right)$
20: **end for**

---

# C  Complexity Analysis

In this section, we conduct a comparative analysis of computational and communication efficiency between FedLASE and SOTA methods (HeteroFL [9], FedRolex [18], ScaleFL [25], and FIARSE [21]), focusing specifically on per-round cost analysis as summarized in Table 5.

**Computational Complexity.** The computational complexity arises from both server-side and client-side operations. On the server side, three primary tasks contribute to the computational load: parameter aggregation, mask computation, and submodel extraction. While all compared methods share the common $\mathcal{O}(d)$ complexity for aggregation and submodel extraction, their mask computation approaches differ in implementation paradigms. HeteroFL, FedRolex, and ScaleFL employ predefined submodel extraction schemes with constant-time mask computation ($\mathcal{O}(1)$). Notably, ScaleFL needs additional computational overhead from solving an optimization subproblem during initialization to determine client-specific width and depth configurations. In comparison, FIARSE and FedLASE require parameter importance evaluation ($\mathcal{O}(d)$) followed by parameter sorting. The global sorting of FIARSE results in $\mathcal{O}(d \log(d))$ complexity, whereas the layer-wise sorting of FedLASE achieves $\mathcal{O}(\sum_{i=1}^{L} d_{l_i} \log(d_{l_i}))$. Given that $d_{l_i} \ll d$ for typical deep learning architectures, our method possesses superior computational efficiency in sorting operations.

Client-side computations involve three core components: loss calculation, gradient computation, and model updating. HeteroFL, FedRolex, FIARSE, and FedLASE have equivalent training *loss computation* complexity (denoted by $\mathcal{O}(C_1)$), while ScaleFL needs an additional cost (denoted by $\mathcal{O}(C_2)$) due to self-distillation, making the total loss computation complexity of $\mathcal{O}(C_1) + \mathcal{O}(C_2)$. Suppose the *gradient computation* of training loss across all methods is $\mathcal{O}(C_2)$. In comparison, ScaleFL introduces an extra cost for gradient calculation due to self-distillation, represented as $\mathcal{O}(C_4)$. The additional gradient computational cost for FIARSE and FedLASE introduced by the STE technique is $\mathcal{O}(d^n)$ with $d^n = r^n d$. For the *model updating*, the computational costs for all approaches are $\mathcal{O}(d^n)$. Therefore, in the local calculation process, our method does not introduce a large amount of computational overhead.

Table 5: Computational and communication complexity comparison per training round

| Method | Computational cost | | | | | | Communication cost | |
|---|---|---|---|---|---|---|---|---|
| | Server | | | Client $n$ | | | Upstream | Downstream |
| | Aggregation | Mask | Submodel extracting | Local loss | Local gradient | Model updating | | |
| HeteroFL | $\mathcal{O}(d)$ | $\mathcal{O}(1)$ | $\mathcal{O}(d)$ | $\mathcal{O}(C_1)$ | $\mathcal{O}(C_2)$ | $\mathcal{O}(d^n)$ | $\mathcal{O}(d^n)$ | $\mathcal{O}(d^n)$ |
| FedRolex | $\mathcal{O}(d)$ | $\mathcal{O}(1)$ | $\mathcal{O}(d)$ | $\mathcal{O}(C_1)$ | $\mathcal{O}(C_2)$ | $\mathcal{O}(d^n)$ | $\mathcal{O}(d^n)$ | $\mathcal{O}(d^n)$ |
| ScaleFL | $\mathcal{O}(d)$ | $\mathcal{O}(1)$ | $\mathcal{O}(d)$ | $\mathcal{O}(C_1) + \mathcal{O}(C_3)$ | $\mathcal{O}(C_2) + \mathcal{O}(C_4)$ | $\mathcal{O}(d^n)$ | $\mathcal{O}(d^n)$ | $\mathcal{O}(d^n)$ |
| FIARSE | $\mathcal{O}(d)$ | $\mathcal{O}(d) + \mathcal{O}(d\log(d))$ | $\mathcal{O}(d)$ | $\mathcal{O}(C_1)$ | $\mathcal{O}(C_2) + \mathcal{O}(d^n)$ | $\mathcal{O}(d^n)$ | $\mathcal{O}(d^n)$ | $\mathcal{O}(d^n)$ |
| FedLASE | $\mathcal{O}(d)$ | $\mathcal{O}(d) + \mathcal{O}(\sum_{i=1}^{L} d_{l_i} \log(d_{l_i}))$ | $\mathcal{O}(d)$ | $\mathcal{O}(C_1)$ | $\mathcal{O}(C_2) + \mathcal{O}(d^n)$ | $\mathcal{O}(d^n)$ | $\mathcal{O}(d^n)$ | $\mathcal{O}(d^n)$ |

\* $d^n = r^n d$; $C_1$: Training loss computation; $C_2$: Gradient computation for training loss; $C_3$ & $C_4$: Self-distillation costs for ScaleFL.

**Communication Complexity.** In terms of communication overhead, all compared methods exhibit equivalent complexity for bidirectional transmission of client submodels ($\mathcal{O}(d^n)$). Although FIARSE and FedLASE require additional threshold communication for submodel extraction, this supplementary cost becomes negligible relative to the dominant model parameter transmission.

Through systematic complexity analysis, we demonstrate that FedLASE achieves a balanced computational and communication complexity. The proposed layer-wise sorting mechanism reduces server-side computation compared to global sorting approaches while maintaining client-side complexity comparable to baseline methods.

# D  Standard Assumptions and Proof of Theorem 1

## D.1  Assumptions

To analyze the convergence of federated learning, the following standard assumptions are commonly used in previous works [19, 22, 21], where Assumptions 2-3 ensure that the gradients are smooth and bounded, and Assumptions 4-5 account for the noise in gradients.

*Assumption 2:* (L-smoothness) The local objective function $\tilde{F}_n(\theta)$ is $L$-smooth, i.e., for any $\theta, \theta' \in \mathbb{R}^d$ and $n$,

$$\|\nabla \tilde{F}_n(\theta) - \nabla \tilde{F}_n(\theta')\| \leq L\|\theta - \theta'\|.$$

*Assumption 3:* (Bounded Gradient) The expected squared norm of the stochastic gradient is bounded uniformly, i.e., for a constant $G > 0$ and any $n, t, k$,

$$\mathbb{E}\|\nabla \tilde{F}_n(\theta_{t,k}^n, \xi_{t,k})\|^2 \leq G.$$

*Assumption 4:* (Gradient Noise for IID Data) For IID data distribution, assume that

$$\mathbb{E}[\nabla \tilde{F}_n(\theta_{t,k}^n, \xi_{t,k})] = \nabla \tilde{F}(\theta_{t,k}^n),$$

and

$$\mathbb{E}\|\nabla \tilde{F}_n(\theta_{t,k}^n, \xi_{t,k}) - \nabla \tilde{F}_n(\theta_{t,k}^n)\|^2 \leq \sigma^2.$$

*Assumption 5:* (Gradient Noise for non-IID Data) For non-IID data distribution, assume that

$$\mathbb{E}\Big[\frac{1}{|\mathcal{N}_{t,i}|} \sum_{n \in \mathcal{N}_{t,i}} \big(\nabla \tilde{F}_n(\theta_{t,k}^n, \xi_{t,k})\big)_i\Big] = \big(\nabla \tilde{F}(\theta_{t,k}^n)\big)_i,$$

and

$$\mathbb{E}\Big\|\frac{1}{|\mathcal{N}_{t,i}|} \sum_{n \in \mathcal{N}_{t,i}} \big(\nabla \tilde{F}_n(\theta_{t,k}^n, \xi_{t,k}) - \nabla \tilde{F}(\theta_{t,k}^n)\big)_i\Big\|^2 \leq \sigma^2,$$

where $\mathcal{N}_{t,i} \triangleq \{n | m_{t,i}^n \geq 1\}$ is the set of clients training the $i$th parameter in round $t$, $m_{t,i}^n$ is the $i$th elements of the mask $M_t^n$ for client $n$ in round $t$, and $|\mathcal{N}_{t,i}|$ is the number of elements in the set $\mathcal{N}_{t,i}$.

## D.2  Proof of Theorem 1

From Assumption 2, we can obtain

$$\mathbb{E}[\tilde{F}(\theta_{t+1})] - \mathbb{E}[\tilde{F}(\theta_t)] \leq \mathbb{E}\left\langle \nabla \tilde{F}(\theta_t), \theta_{t+1} - \theta_t \right\rangle + \frac{L}{2}\mathbb{E}\|\theta_{t+1} - \theta_t\|^2. \tag{6}$$

In the sequel, we analyze the upper bounds of each term on the right side of Eq. (6). Before this, we first calculate the difference between the global models at $t+1$th round and $t$th round. Let

552 $\mathcal{N}_{t,i} \triangleq \{n|m_{t,i}^n \geq 1\}$ denote the set of clients training the $i$th parameter in round $t$. Then for the $i$th
553 element of the global model ($i \in \mathcal{I}_t \triangleq \{i| \sum_{n=1}^N m_{t,i}^n \geq 1\}$), we have

$$
\begin{aligned}
\theta_{t+1,i} - \theta_{t,i} &= \left( \frac{1}{|\mathcal{N}_{t,i}|} \sum_{n \in \mathcal{N}_{t,i}} \theta_{t,K,i}^n \right) - \theta_{t,i} \\
&\overset{(a)}{=} \frac{1}{|\mathcal{N}_{t,i}|} \sum_{n \in \mathcal{N}_{t,i}} \left[ \theta_{t,i} \cdot m_{t,i}^n - \sum_{k=0}^{K-1} \eta \left( \nabla_{\theta_{t,k}^n} \tilde{F}_n(\theta_{t,k}^n \odot M_t^n, \xi_{t,k}^n) \right)_i \right] - \theta_{t,i} \quad (7) \\
&\overset{(b)}{=} -\frac{1}{|\mathcal{N}_{t,i}|} \sum_{n \in \mathcal{N}_{t,i}} \sum_{k=0}^{K-1} \eta \left( \nabla_{\theta_{t,k}^n} \tilde{F}_n(\theta_{t,k}^n \odot M_t^n, \xi_{t,k}^n) \right)_i,
\end{aligned}
$$

554 where $m_{t,i}^n$ is the $i$th elements of the mask $M_t^n$ for client $n$ in round $t$, (a) is obtained by the global
555 aggregation $\theta_{t+1} = \frac{\sum_n M_t^n \odot \theta_t^n}{\sum_n M_t^n} = \frac{\sum_n M_t^n \odot \theta_{t,K}^n}{\sum_n M_t^n}$ and local training $\theta_{t,k+1}^n = \theta_{t,k}^n - \nabla_{\theta_{t,k}^n} \tilde{F}_n(\theta_{t,k}^n \odot$
556 $M_t^n, \xi_{t,k}^n)$, and (b) holds because $m_{t,i}^n = 1$ when $n \in \mathcal{N}_{t,i}$.

557 The first term on the right side of Eq. (6) can be amplified as

$$
\begin{aligned}
&\mathbb{E} \left\langle \nabla \tilde{F}(\theta_t), \theta_{t+1} - \theta_t \right\rangle \\
&= \sum_{i \in \mathcal{I}_t} \mathbb{E} \left[ (\nabla \tilde{F}(\theta_t))_i \cdot (\theta_{t+1,i} - \theta_{t,i}) \right] \\
&\overset{(a)}{=} \sum_{i \in \mathcal{I}_t} \mathbb{E} \left[ (\nabla \tilde{F}(\theta_t))_i \cdot \left( -\frac{1}{|\mathcal{N}_{t,i}|} \sum_{n \in \mathcal{N}_{t,i}} \sum_{k=0}^{K-1} \eta \left( \nabla_{\theta_{t,k}^n} \tilde{F}_n(\theta_{t,k}^n \odot M_t^n, \xi_{t,k}^n) \right)_i \right) \right] \\
&= -\eta K \sum_{i \in \mathcal{I}_t} \mathbb{E}(\nabla \tilde{F}(\theta_t))_i^2 - \sum_{i \in \mathcal{I}_t} \mathbb{E} \left[ (\nabla \tilde{F}(\theta_t))_i \cdot \left( \frac{1}{|\mathcal{N}_{t,i}|} \sum_{n \in \mathcal{N}_{t,i}} \sum_{k=0}^{K-1} \eta \left( \nabla_{\theta_{t,k}^n} \tilde{F}_n(\theta_{t,k}^n \odot M_t^n, \xi_{t,k}^n) - \nabla \tilde{F}(\theta_t) \right)_i \right) \right] \\
&= -\eta K \sum_{i \in \mathcal{I}_t} \mathbb{E}(\nabla \tilde{F}(\theta_t))_i^2 - \eta K \sum_{i \in \mathcal{I}_t} \mathbb{E} \left[ (\nabla \tilde{F}(\theta_t))_i \cdot \left( \frac{1}{K|\mathcal{N}_{t,i}|} \sum_{n \in \mathcal{N}_{t,i}} \sum_{k=0}^{K-1} \left( \nabla_{\theta_{t,k}^n} \tilde{F}_n(\theta_{t,k}^n \odot M_t^n, \xi_{t,k}^n) - \nabla \tilde{F}(\theta_t) \right)_i \right) \right] \\
&\overset{(b)}{\leq} -\eta K \sum_{i \in \mathcal{I}_t} \mathbb{E}(\nabla \tilde{F}(\theta_t))_i^2 + \frac{\eta K}{2} \sum_{i \in \mathcal{I}_t} \mathbb{E} \left[ (\nabla \tilde{F}(\theta_t))_i \right]^2 \\
&\quad + \frac{\eta K}{2} \sum_{i \in \mathcal{I}_t} \mathbb{E} \left[ \left( \frac{1}{K|\mathcal{N}_{t,i}|} \sum_{n \in \mathcal{N}_{t,i}} \sum_{k=0}^{K-1} \left( \nabla_{\theta_{t,k}^n} \tilde{F}_n(\theta_{t,k}^n \odot M_t^n, \xi_{t,k}^n) - \nabla \tilde{F}(\theta_t) \right)_i \right) \right]^2,
\end{aligned}
$$

$$(8)$$

558 where (a) comes from Eq. (7), (b) holds because $ab \leq \frac{1}{2}(a^2 + b^2)$. The third term on the right side of
559 Eq. (8) is bounded by

$$
\begin{aligned}
&\frac{\eta K}{2} \sum_{i \in \mathcal{I}_t} \mathbb{E} \left[ \left( \frac{1}{K|\mathcal{N}_{t,i}|} \sum_{n \in \mathcal{N}_{t,i}} \sum_{k=0}^{K-1} \left( \nabla_{\theta_{t,k}^n} \tilde{F}_n(\theta_{t,k}^n \odot M_t^n, \xi_{t,k}^n) - \nabla \tilde{F}(\theta_t) \right)_i \right) \right]^2 \\
&\overset{(a)}{\leq} \frac{\eta K}{2} \sum_{i \in \mathcal{I}_t} \frac{1}{K|\mathcal{N}_{t,i}|} \sum_{n \in \mathcal{N}_{t,i}} \sum_{k=0}^{K-1} \mathbb{E} \left[ \left( \nabla_{\theta_{t,k}^n} \tilde{F}_n(\theta_{t,k}^n \odot M_t^n, \xi_{t,k}^n) - \nabla \tilde{F}_n(\theta_t, \xi_t) \right)_i \right]^2 \\
&\leq \frac{\eta K}{2} \frac{1}{K|\mathcal{N}_{t,i}|_{\min}} \sum_{n=1}^N \sum_{k=0}^{K-1} \mathbb{E} \left\| \nabla_{\theta_{t,k}^n} \tilde{F}_n(\theta_{t,k}^n \odot M_t^n, \xi_{t,k}^n) - \nabla \tilde{F}_n(\theta_t, \xi_t) \right\|^2 \\
&\overset{(b)}{\leq} \frac{\eta K}{2} \frac{L^2}{K|\mathcal{N}_{t,i}|_{\min}} \sum_{n=1}^N \sum_{k=0}^{K-1} \mathbb{E} \left\| \theta_{t,k}^n \odot M_t^n - \theta_t \right\|^2,
\end{aligned}
$$

where $|\mathcal{N}_{t,i}|_{\min} = \min_i\{|\mathcal{N}_{t,i}|\}$, (a) holds because $\|\frac{1}{s}\sum_{i=1}^s a_i\|^2 \leq \frac{1}{s}\sum_{i=1}^s \|a_i\|^2$, and (b) comes from Assumption 2. By introducing an additional term $\theta_{t,0}^n \odot M_t^n$, the above inequality can be further amplified as

$$
\frac{\eta K}{2}\sum_{i\in\mathcal{I}_t}\mathbb{E}\left[\left(\frac{1}{K|\mathcal{N}_{t,i}|}\sum_{n\in\mathcal{N}_{t,i}}\sum_{k=0}^{K-1}\left(\nabla_{\theta_{t,k}^n}\tilde{F}_n(\theta_{t,k}^n\odot M_t^n,\xi_{t,k}^n)-\nabla\tilde{F}(\theta_t)\right)_i\right)\right]^2
$$

$$
\leq\frac{\eta K}{2}\frac{L^2}{K|\mathcal{N}_{t,i}|_{\min}}\sum_{n=1}^N\sum_{k=0}^{K-1}\mathbb{E}\left\|\theta_{t,k}^n\odot M_t^n-\theta_{t,0}^n\odot M_t^n+\theta_t\odot M_t^n-\theta_t\right\|^2
$$

$$
\leq\frac{\eta K}{2}\frac{2L^2}{K|\mathcal{N}_{t,i}|_{\min}}K\sum_{n=1}^N\mathbb{E}\left\|\theta_t\odot M_t^n-\theta_t\right\|^2+\frac{\eta K}{2}\frac{2L^2}{K|\mathcal{N}_{t,i}|_{\min}}\sum_{n=1}^N\sum_{k=0}^{K-1}\mathbb{E}\left\|\theta_{t,k}^n\odot M_t^n-\theta_{t,0}^n\odot M_t^n\right\|^2
$$

$$
\overset{(a)}{=}\frac{\eta K}{2}\frac{2L^2}{K|\mathcal{N}_{t,i}|_{\min}}K\sum_{n=1}^N\mathbb{E}\left\|\theta_t\odot M_t^n-\theta_t\right\|^2+\frac{\eta K}{2}\frac{2L^2}{K|\mathcal{N}_{t,i}|_{\min}}\sum_{n=1}^N\sum_{k=1}^{K-1}\mathbb{E}\left\|-\sum_{j=0}^{k-1}\eta\nabla_{\theta_{t,j}^n}\tilde{F}_n(\theta_{t,j}^n\odot M_t^n)\right\|^2
$$

$$
\overset{(b)}{\leq}\frac{\eta K}{2}\frac{2L^2}{K|\mathcal{N}_{t,i}|_{\min}}K\sum_{n=1}^N\mathbb{E}\left\|\theta_t\odot M_t^n-\theta_t\right\|^2+\frac{\eta K}{2}\frac{2L^2}{K|\mathcal{N}_{t,i}|_{\min}}\sum_{n=1}^N\sum_{k=1}^{K-1}k\sum_{j=0}^{k-1}\mathbb{E}\left\|-\eta\nabla_{\theta_{t,j}^n}\tilde{F}_n(\theta_{t,j}^n\odot M_t^n)\right\|^2,
$$
$$\tag{9}$$

where (a) is obtained by the local updates, (b) holds because $\|\sum_{i=1}^s a_i\|^2 \leq s\sum_{i=1}^s\|a_i\|^2$.

Combining Eqs. (9) with Eq. (8) gives

$$
\mathbb{E}\left\langle\nabla\tilde{F}(\theta_t),\theta_{t+1}-\theta_t\right\rangle
$$

$$
\leq-\eta K\sum_{i\in\mathcal{I}_t}\mathbb{E}(\nabla\tilde{F}(\theta_t))_i^2+\frac{\eta K}{2}\sum_{i\in\mathcal{I}_t}\mathbb{E}\left[(\nabla\tilde{F}(\theta_t))_i\right]^2
$$

$$
+\frac{\eta K}{2}\sum_{i\in\mathcal{I}_t}\mathbb{E}\left[\left(\frac{1}{K|\mathcal{N}_{t,i}|}\sum_{n\in\mathcal{N}_{t,i}}\sum_{k=0}^{K-1}\left(\nabla_{\theta_{t,k}^n}\tilde{F}_n(\theta_{t,k}^n\odot M_t^n,\xi_{t,k}^n)-\nabla\tilde{F}(\theta_t)\right)_i\right)\right]^2
$$

$$
\leq-\tfrac{\eta K}{2}\sum_{i\in\mathcal{I}_t}\mathbb{E}\left[(\nabla\tilde{F}(\theta_t))_i\right]^2+\tfrac{\eta K}{2}\tfrac{2L^2}{K|\mathcal{N}_{t,i}|_{\min}}K\sum_{n=1}^N\mathbb{E}\left\|\theta_t\odot M_t^n-\theta_t\right\|^2+\tfrac{\eta K}{2}\tfrac{2L^2}{K|\mathcal{N}_{t,i}|_{\min}}\sum_{n=1}^N\sum_{k=1}^{K-1}k\sum_{j=0}^{k-1}\mathbb{E}\left\|-\eta\nabla_{\theta_{t,j}^n}\tilde{F}_n(\theta_{t,j}^n\odot M_t^n)\right\|^2.
$$
$$\tag{10}$$

For another term on the right side of Eq. (6), we have

$$
\frac{L}{2}\mathbb{E}\|\theta_{t+1}-\theta_t\|^2
$$

$$
\overset{(a)}{=}\frac{L}{2}\sum_{i\in\mathcal{I}_t}\mathbb{E}\left[-\frac{1}{|\mathcal{N}_{t,i}|}\sum_{n\in\mathcal{N}_{t,i}}\sum_{k=0}^{K-1}\eta\left(\nabla_{\theta_{t,k}^n}\tilde{F}_n(\theta_{t,k}^n\odot M_t^n,\xi_{t,k}^n)\right)_i\right]^2
$$

$$
=\frac{L}{2}\sum_{i\in\mathcal{I}_t}\mathbb{E}\left[\frac{1}{|\mathcal{N}_{t,i}|}\sum_{n\in\mathcal{N}_{t,i}}\sum_{k=0}^{K-1}\eta\left(\nabla_{\theta_{t,k}^n}\tilde{F}_n(\theta_{t,k}^n\odot M_t^n,\xi_{t,k}^n)-\nabla_{\theta_{t,k}^n}\tilde{F}_n(\theta_{t,k}^n\odot M_t^n)\right)_i\right.
$$

$$
\left.+\frac{1}{|\mathcal{N}_{t,i}|}\sum_{n\in\mathcal{N}_{t,i}}\sum_{k=0}^{K-1}\eta\left(\nabla_{\theta_{t,k}^n}\tilde{F}_n(\theta_{t,k}^n\odot M_t^n)-\nabla\tilde{F}_n(\theta_t)\right)_i+\frac{1}{|\mathcal{N}_{t,i}|}\sum_{n\in\mathcal{N}_{t,i}}\sum_{k=0}^{K-1}\eta\left(\nabla\tilde{F}_n(\theta_t)\right)_i\right]^2
$$

$$
\overset{(b)}{\leq}\frac{3L}{2}\sum_{i\in\mathcal{I}_t}\mathbb{E}\left[\frac{1}{|\mathcal{N}_{t,i}|}\sum_{n\in\mathcal{N}_{t,i}}\sum_{k=0}^{K-1}\eta\left(\nabla_{\theta_{t,k}^n}\tilde{F}_n(\theta_{t,k}^n\odot M_t^n,\xi_{t,k}^n)-\nabla_{\theta_{t,k}^n}\tilde{F}_n(\theta_{t,k}^n\odot M_t^n)\right)_i\right]^2
$$

$$
+\frac{3L}{2}\sum_{i\in\mathcal{I}_t}\mathbb{E}\left[\frac{1}{|\mathcal{N}_{t,i}|}\sum_{n\in\mathcal{N}_{t,i}}\sum_{k=0}^{K-1}\eta\left(\nabla_{\theta_{t,k}^n}\tilde{F}_n(\theta_{t,k}^n\odot M_t^n)-\nabla\tilde{F}_n(\theta_t)\right)_i\right]^2+\frac{3\eta^2K^2L}{2}\sum_{i\in\mathcal{I}_t}\mathbb{E}\left(\nabla\tilde{F}(\theta_t)\right)_i^2,
$$
$$\tag{11}$$

where (a) comes from Eq. (7), (b) is obtained by $\|\sum_{i=1}^{s} a_i\|^2 \le s \sum_{i=1}^{s} \|a_i\|^2$ and Assumptions 4 and 5. Combining Assumption 2 and Eq. (9), the second term on the right side of the above inequality can be amplified as

$$
\begin{aligned}
&\frac{3L}{2} \mathbb{E} \sum_{i \in \mathcal{I}_t} \left[ \frac{1}{|\mathcal{N}_{t,i}|} \sum_{n \in \mathcal{N}_{t,i}} \sum_{k=0}^{K-1} \eta \left( \nabla_{\theta_{t,k}^n} \tilde{F}_n(\theta_{t,k}^n \odot M_t^n) - \nabla \tilde{F}_n(\theta_t) \right)_i \right]^2 \\
&\le \frac{3\eta^2 K^2 L}{2} \frac{1}{K|\mathcal{N}_{t,i}|_{\min}} \sum_{n=1}^{N} \sum_{k=0}^{K-1} \mathbb{E} \left\| \nabla_{\theta_{t,k}^n} \tilde{F}_n(\theta_{t,k}^n \odot M_t^n) - \nabla \tilde{F}_n(\theta_t) \right\|^2 \\
&\le \frac{3\eta^2 K^2 L}{2} \frac{L^2}{K|\mathcal{N}_{t,i}|_{\min}} \sum_{n=1}^{N} \sum_{k=0}^{K-1} \mathbb{E} \left\| \theta_{t,k}^n \odot M_t^n - \theta_t \right\|^2 \\
&\le \frac{3\eta^2 K^2 L}{2} \frac{2KL^2}{K|\mathcal{N}_{t,i}|_{\min}} \sum_{n=1}^{N} \mathbb{E} \left\| \theta_t \odot M_t^n - \theta_t \right\|^2 \\
&\quad + \frac{3\eta^2 K^2 L}{2} \frac{2L^2}{K|\mathcal{N}_{t,i}|_{\min}} \sum_{n=1}^{N} \sum_{k=1}^{K-1} k \sum_{j=0}^{k-1} \mathbb{E} \left\| - \eta \nabla_{\theta_{t,j}^n} \tilde{F}_n(\theta_{t,j}^n \odot M_t^n) \right\|^2 .
\end{aligned}
\tag{12}
$$

According to Assumptions 4 and 5, the first term on the right side of Eq. (11) is bounded by

i) iid

$$
\begin{aligned}
&\frac{3L}{2} \sum_{i \in \mathcal{I}_t} \mathbb{E} \left[ \frac{1}{|\mathcal{N}_{t,i}|} \sum_{n \in \mathcal{N}_{t,i}} \sum_{k=0}^{K-1} \eta \left( \nabla_{\theta_{t,k}^n} \tilde{F}_n(\theta_{t,k}^n \odot M_t^n, \xi_{t,k}^n) - \nabla_{\theta_{t,k}^n} \tilde{F}_n(\theta_{t,k}^n \odot M_t^n) \right)_i \right]^2 \\
&= \frac{3\eta^2 K^2 L}{2} \sum_{i \in \mathcal{I}_t} \mathbb{E} \left[ \frac{1}{K|\mathcal{N}_{t,i}|} \sum_{n \in \mathcal{N}_{t,i}} \sum_{k=0}^{K-1} \left( \nabla_{\theta_{t,k}^n} \tilde{F}_n(\theta_{t,k}^n \odot M_t^n, \xi_{t,k}^n) - \nabla_{\theta_{t,k}^n} \tilde{F}_n(\theta_{t,k}^n \odot M_t^n) \right)_i \right]^2 \\
&\le \frac{3\eta^2 K^2 L}{2} \sum_{i \in \mathcal{I}_t} \frac{1}{K|\mathcal{N}_{t,i}|} \sum_{n \in \mathcal{N}_{t,i}} \sum_{k=0}^{K-1} \mathbb{E} \left[ \left( \nabla_{\theta_{t,k}^n} \tilde{F}_n(\theta_{t,k}^n \odot M_t^n, \xi_{t,k}^n) - \nabla_{\theta_{t,k}^n} \tilde{F}_n(\theta_{t,k}^n \odot M_t^n) \right)_i \right]^2 \\
&\le \frac{3\eta^2 K^2 L}{2} \frac{1}{K|\mathcal{N}_{t,i}|_{\min}} \sum_{n=1}^{N} \sum_{k=0}^{K-1} \mathbb{E} \left\| \nabla_{\theta_{t,k}^n} \tilde{F}_n(\theta_{t,k}^n \odot M_t^n, \xi_{t,k}^n) - \nabla_{\theta_{t,k}^n} \tilde{F}_n(\theta_{t,k}^n \odot M_t^n) \right\|^2 \\
&\le \frac{3\eta^2 K^2 L}{2} \frac{NK\sigma^2}{K|\mathcal{N}_{t,i}|_{\min}}
\end{aligned}
\tag{13}
$$

ii) non-iid

$$
\begin{aligned}
&\frac{3L}{2} \sum_{i \in \mathcal{I}_t} \mathbb{E} \left[ \frac{1}{|\mathcal{N}_{t,i}|} \sum_{n \in \mathcal{N}_{t,i}} \sum_{k=0}^{K-1} \eta \left( \nabla_{\theta_{t,k}^n} \tilde{F}_n(\theta_{t,k}^n \odot M_t^n, \xi_{t,k}^n) - \nabla_{\theta_{t,k}^n} \tilde{F}_n(\theta_{t,k}^n \odot M_t^n) \right)_i \right]^2 \\
&= \frac{3\eta^2 K^2 L}{2} \sum_{i \in \mathcal{I}_t} \mathbb{E} \left[ \frac{1}{K} \sum_{k=0}^{K-1} \frac{1}{|\mathcal{N}_{t,i}|} \sum_{n \in \mathcal{N}_{t,i}} \left( \nabla_{\theta_{t,k}^n} \tilde{F}_n(\theta_{t,k}^n \odot M_t^n, \xi_{t,k}^n) - \nabla_{\theta_{t,k}^n} \tilde{F}_n(\theta_{t,k}^n \odot M_t^n) \right)_i \right]^2 \\
&\le \frac{3\eta^2 K^2 L}{2} \sum_{i \in \mathcal{I}_t} \frac{1}{K} \sum_{k=0}^{K-1} \mathbb{E} \left[ \frac{1}{|\mathcal{N}_{t,i}|} \sum_{n \in \mathcal{N}_{t,i}} \left( \nabla_{\theta_{t,k}^n} \tilde{F}_n(\theta_{t,k}^n \odot M_t^n, \xi_{t,k}^n) - \nabla_{\theta_{t,k}^n} \tilde{F}_n(\theta_{t,k}^n \odot M_t^n) \right)_i \right]^2 \\
&\le \frac{3\eta^2 K^2 L \sigma^2 d}{2}
\end{aligned}
\tag{14}
$$

572 Substituting Eqs. (12)-(14) into Eq. (11), we have

$$
\frac{L}{2}\mathbb{E}\|\theta_{t+1}-\theta_t\|^2
$$

$$
\leq \frac{3L}{2}\mathbb{E}\sum_{i\in\mathcal{I}_t}\left[\frac{1}{|\mathcal{N}_{t,i}|}\sum_{n\in\mathcal{N}_{t,i}}\sum_{k=0}^{K-1}\eta\Big(\nabla_{\theta_{t,k}^n}\tilde{F}_n(\theta_{t,k}^n\odot M_t^n,\xi_{t,k}^n)-\nabla_{\theta_{t,k}^n}\tilde{F}_n(\theta_{t,k}^n\odot M_t^n)\Big)_i\right]^2
$$

$$
+\frac{3L}{2}\mathbb{E}\sum_{i\in\mathcal{I}_t}\left[\frac{1}{|\mathcal{N}_{t,i}|}\sum_{n\in\mathcal{N}_{t,i}}\sum_{k=0}^{K-1}\eta\Big(\nabla_{\theta_{t,k}^n}\tilde{F}_n(\theta_{t,k}^n\odot M_t^n)-\nabla\tilde{F}_n(\theta_t)\Big)_i\right]^2+\frac{3\eta^2K^2L}{2}\mathbb{E}\sum_{i\in\mathcal{I}_t}\Big(\nabla\tilde{F}(\theta_t)\Big)_i^2
$$

$$
\leq\frac{3\eta^2K^2L}{2}\mathbb{E}\sum_{i\in\mathcal{I}_t}\Big(\nabla\tilde{F}(\theta_t)\Big)_i^2+\frac{3\eta^2K^2L}{2}\frac{NK\sigma^2}{K|\mathcal{N}_{t,i}|_{\min}}\quad\text{(iid)}+\frac{3\eta^2K^2L\sigma^2d}{2}\quad\text{(non-iid)}
$$

$$
+\frac{3\eta^2K^2L}{2}\frac{2KL^2}{K|\mathcal{N}_{t,i}|_{\min}}\sum_{n=1}^N\mathbb{E}\Big\|\theta_t\odot M_t^n-\theta_t\Big\|^2+\frac{3\eta^2K^2L}{2}\frac{2L^2}{K|\mathcal{N}_{t,i}|_{\min}}\sum_{n=1}^N\sum_{k=1}^{K-1}k\sum_{j=0}^{k-1}\mathbb{E}\Big\|-\eta\nabla_{\theta_{t,j}^n}\tilde{F}_n(\theta_{t,j}^n\odot M_t^n)\Big\|^2
$$

(15)

573 From Eqs. (6), (10), and (15), one can get

$$
\mathbb{E}[\tilde{F}(\theta_{t+1})]-\mathbb{E}[\tilde{F}(\theta_t)]
$$

$$
\leq\mathbb{E}\Big\langle\nabla\tilde{F}(\theta_t),\theta_{t+1}-\theta_t\Big\rangle+\frac{L}{2}\mathbb{E}\|\theta_{t+1}-\theta_t\|^2
$$

$$
\leq\frac{\eta K}{2}\frac{2L^2}{K|\mathcal{N}_{t,i}|_{\min}}K\sum_{n=1}^N\mathbb{E}\Big\|\theta_t\odot M_t^n-\theta_t\Big\|^2+\frac{\eta K}{2}\frac{2L^2}{K|\mathcal{N}_{t,i}|_{\min}}\sum_{n=1}^N\sum_{k=1}^{K-1}k\sum_{j=0}^{k-1}\mathbb{E}\Big\|-\eta\nabla_{\theta_{t,j}^n}\tilde{F}_n(\theta_{t,j}^n\odot M_t^n)\Big\|^2
$$

$$
-\frac{\eta K}{2}\sum_{i\in\mathcal{I}_t}\mathbb{E}\left[(\nabla\tilde{F}(\theta_t))_i\right]^2+\frac{3\eta^2K^2L}{2}\mathbb{E}\sum_{i\in\mathcal{I}_t}\Big(\nabla\tilde{F}(\theta_t)\Big)_i^2
$$

$$
+\frac{3\eta^2K^2L}{2}\frac{NK\sigma^2}{K|\mathcal{N}_{t,i}|_{\min}}\quad\text{(iid)}+\frac{3\eta^2K^2L\sigma^2d}{2}\quad\text{(non-iid)}+\frac{3\eta^2K^2L}{2}\frac{2KL^2}{K|\mathcal{N}_{t,i}|_{\min}}\sum_{n=1}^N\mathbb{E}\Big\|\theta_t\odot M_t^n-\theta_t\Big\|^2
$$

$$
+\frac{3\eta^2K^2L}{2}\frac{2L^2}{K|\mathcal{N}_{t,i}|_{\min}}\sum_{n=1}^N\sum_{k=1}^{K-1}k\sum_{j=0}^{k-1}\mathbb{E}\Big\|-\eta\nabla_{\theta_{t,j}^n}\tilde{F}_n(\theta_{t,j}^n\odot M_t^n)\Big\|^2
$$

$$
=\left(-\frac{\eta K}{2}+\frac{3\eta^2K^2L}{2}\right)\sum_{i\in\mathcal{I}_t}\mathbb{E}\Big(\nabla\tilde{F}(\theta_t)\Big)_i^2+\frac{3\eta^2K^2L}{2}\frac{NK\sigma^2}{K|\mathcal{N}_{t,i}|_{\min}}\quad\text{(iid)}+\frac{3\eta^2K^2L\sigma^2d}{2}\quad\text{(non-iid)}
$$

$$
+\left(\frac{\eta K}{2}\frac{2L^2}{K|\mathcal{N}_{t,i}|_{\min}}K+\frac{3\eta^2K^2L}{2}\frac{2KL^2}{K|\mathcal{N}_{t,i}|_{\min}}\right)\sum_{n=1}^N\mathbb{E}\Big\|\theta_t\odot M_t^n-\theta_t\Big\|^2
$$

$$
+\left(\frac{\eta K}{2}\frac{2L^2}{K|\mathcal{N}_{t,i}|_{\min}}+\frac{3\eta^2K^2L}{2}\frac{2L^2}{K|\mathcal{N}_{t,i}|_{\min}}\right)\sum_{n=1}^N\sum_{k=1}^{K-1}k\sum_{j=0}^{k-1}\mathbb{E}\Big\|-\eta\nabla_{\theta_{t,j}^n}\tilde{F}_n(\theta_{t,j}^n\odot M_t^n)\Big\|^2
$$

$$
\overset{(a)}{\leq}-\frac{\eta K[1-(1-6\eta KL)]}{4}\sum_{i\in\mathcal{I}_t}\mathbb{E}\Big(\nabla\tilde{F}(\theta_t)\Big)_i^2+\frac{3\eta^2K^2L}{2}\frac{NK\sigma^2}{K|\mathcal{N}_{t,i}|_{\min}}\quad\text{(iid)}+\frac{3\eta^2K^2L\sigma^2d}{2}\quad\text{(non-iid)}
$$

$$
+\frac{\eta KL^2(1+3\eta KL)}{|\mathcal{N}_{t,i}|_{\min}}\sum_{i=1}^p N_i(1-\text{level}_i)\delta_i^2\|\theta_t\|^2+\frac{\eta KL^2(1+3\eta L)}{|\mathcal{N}_{t,i}|_{\min}}\sum_{n=1}^N\sum_{k=1}^{K-1}k\sum_{j=0}^{k-1}\eta^2G
$$

$$
\overset{(b)}{\leq}-\frac{\eta K}{4}\sum_{i\in\mathcal{I}_t}\mathbb{E}\Big(\nabla\tilde{F}(\theta_t)\Big)_i^2+\frac{3\eta^2K^2LN\sigma^2}{2|\mathcal{N}_{t,i}|_{\min}}\quad\text{(iid)}+\frac{3\eta^2K^2L\sigma^2d}{2}\quad\text{(non-iid)}
$$

$$
+\frac{\eta KL^2(1+3\eta KL)}{|\mathcal{N}_{t,i}|_{\min}}\sum_{i=1}^p N_i(1-\text{level}_i)\delta_i^2\|\theta_t\|^2+\frac{\eta^3KL^2NG(1+3\eta L)}{|\mathcal{N}_{t,i}|_{\min}}\sum_{k=1}^{K-1}k^2,
$$

(16)

where (a) comes from Assumptions 3 and 1, (b) is given by $6\eta KL < 1$. Taking the sum over $t = 0, 1, \ldots, T - 1$ on both sides of the above inequality gives

$$
\frac{1}{T} \sum_{t=0}^{T-1} \sum_{i \in \mathcal{I}_t} \mathbb{E}\left(\nabla \tilde{F}(\theta_t)\right)_i^2
$$

$$
\leq \frac{4}{\eta KT} \left[ \mathbb{E}[\tilde{F}(\theta_0)] - \mathbb{E}[\tilde{F}(\theta_T)] + \frac{3\eta^2 K^2 LN\sigma^2 T}{2|\mathcal{N}_{t,i}|_{\min}} \quad \text{(iid)} + \frac{3\eta^2 K^2 L\sigma^2 dT}{2} \quad \text{(non-iid)} \right]
$$

$$
+ \frac{4}{\eta KT} \left[ \frac{\eta KL^2(1 + 3\eta KL)}{|\mathcal{N}_{t,i}|_{\min}} \sum_{t=0}^{T-1} \sum_{i=1}^{p} N_i(1 - \text{level}_i)\delta_i^2 \|\theta_t\|^2 + \frac{\eta^3 K^2 L^2 NGT}{6|\mathcal{N}_{t,i}|_{\min}}(1 + 3\eta L)(K - 1)(2K - 1) \right]
$$

$$
\leq \frac{4}{\eta KT} \mathbb{E}[\tilde{F}(\theta_0)] + \frac{6\eta KLN\sigma^2}{|\mathcal{N}_{t,i}|_{\min}} \quad \text{(iid)} + 6\eta KL\sigma^2 d \quad \text{(non-iid)}
$$

$$
+ \frac{4L^2(1 + 3\eta KL)}{|\mathcal{N}_{t,i}|_{\min}T} \sum_{t=0}^{T-1} \sum_{i=1}^{p} N_i(1 - \text{level}_i)\delta_i^2 \|\theta_t\|^2 + \frac{2\eta^2 KL^2 NG}{3|\mathcal{N}_{t,i}|_{\min}}(1 + 3\eta L)(K - 1)(2K - 1)
$$

$$
= \frac{Q_1}{\eta KT} + Q_2 \eta K \quad \text{(iid)} + Q_3 \eta K \quad \text{(non-iid)} + Q_4(1 + 3\eta KL)\frac{1}{T} \sum_{t=0}^{T-1} \sum_{i=1}^{p} N_i(1 - \text{level}_i)\delta_i^2 \|\theta_t\|^2
$$

$$
+ Q_5 \eta^2(1 + 3\eta L)K(K - 1)(2K - 1)
$$

$$
\stackrel{(a)}{=} \mathcal{O}(\frac{1}{\sqrt{T}}) + \mathcal{O}(\frac{1}{\sqrt{T}})\frac{1}{T} \sum_{t=0}^{T-1} \sum_{i=1}^{p} N_i(1 - \text{level}_i)\delta_i^2 \|\theta_t\|^2 + \frac{Q_4}{T} \sum_{t=0}^{T-1} \sum_{i=1}^{p} N_i(1 - \text{level}_i)\delta_i^2 \|\theta_t\|^2,
$$

$$(17)$$

where $Q_1 = 4\mathbb{E}[\tilde{F}(\theta_0)], Q_2 = \frac{6LN\sigma^2}{|\mathcal{N}_{t,i}|_{\min}}, Q_3 = 6L\sigma^2 d, Q_4 = \frac{4L^2}{|\mathcal{N}_{t,i}|_{\min}}, Q_5 = \frac{2L^2 NG}{3|\mathcal{N}_{t,i}|_{\min}}$, (a) holds because $\eta = \mathcal{O}(\frac{1}{K\sqrt{T}})$. This completes the proof.

# E Limitations

In this paper, we propose FedLASE, an importance-aware layer-adaptive submodel extraction framework that selects critical parameters within each layer based on both parameter and layer importance. This design enables structurally consistent and expressive submodels, leading to balanced performance across heterogeneous clients and improved convergence. Although the proposed strategy is effective and computationally efficient, it may not be the theoretically optimal extraction solution. Future work could explore more principled submodel construction methods from an optimization perspective. Nonetheless, the primary objective of this work is to emphasize the importance of assigning appropriate layer-wise extraction ratios for each client in system-heterogeneous federated learning, especially for the case that high-resource clients are few and the majority are resource-constrained.

# F Broader impacts

This work highlights the potential of shifting large-scale model training from centralized computing resources to decentralized collaborative paradigms. With the advancement of federated learning, individual users, small organizations, and resource-constrained devices can increasingly participate in model training, reducing dependence on traditional computing monopolies and improving the accessibility and openness of AI technologies. In particular, our method is well suited for practical deployment scenarios where a few clients have abundant computational resources while most are resource-limited, offering a more feasible solution for real-world applications.

