# OpenReview forum: "FedLASE: Performance-Balanced System-Heterogeneous FL via Layer-Adaptive Submodel Extraction"
_NeurIPS.cc/2025/Conference — Submitted to NeurIPS 2025_

### Official Review · Reviewer_EUdt · 2025-06-20

**Clarity:** 3
**Significance:** 2
**Originality:** 2
**Rating:** 4
**Confidence:** 3

**Summary:**

This paper proposes a method to overcome system heterogeneity and improve performance in Federated Learning (FL). In particular, when clients have varying memory capacities and computational capabilities, ㅣlatency can occur due to differences in computation time between clients. To mitigate this, the paper introduces a method where submodels are extracted and computed according to each client’s capabilities using masks. Clients train their masked models, and the server aggregates the masked updates.

**Questions:**

Looking at Table 1, it is observed that clients with fewer resources tend to achieve better performance. This phenomenon requires further analysis. Could it be due to overfitting?
While the method theoretically improves memory usage and computation speed, I am also curious whether these theoretical benefits can be preserved in practice on GPU systems that process tasks in parallel, specifically, without significant modifications to existing GPU infrastructures.
Despite the existence of [21], please elaborate more on the unique contributions of the proposed method.

**Ethical Concerns:**

["NO or VERY MINOR ethics concerns only"]

**Final Justification:**

The authors provided detailed responses to this reviewer’s questions, including additional experiments. As the concerns of this reviewer have been appropriately discussed, this reviewer would like to keep the initial positive rating.

**Limitations:**

Yes

**Paper Formatting Concerns:**

I don't have any other opinions.

**Quality:**

3

**Strengths And Weaknesses:**

The main idea of the method lies in how the masks are generated. Instead of evaluating the importance of individual parameters, the proposed approach considers the relative importance of entire layers and masks parameters. This prevents any specific layer from excessively losing parameters, thereby preserving the structure and learning ability of important parts of the model.
The proposed method also includes a theoretical analysis, and experimental results show that the proposed method outperforms comparative baselines.
However, the proposed method shares a similar idea with prior work [21]. The STE technique used for model training is also based on prior work. Thus, while the paper has novelty, it is somewhat limited by the contributions of [21].

---

> ### Author Rebuttal · Authors · 2025-07-31
>
> **Response to [Q1]: Clarification on why clients with fewer resources achieve better performance in Table 1.**
>
> We thank the reviewer for pointing out this interesting observation. We provide the following possible explanations based on our analysis and supporting experiments:
>
> - Training dominance of resource-limited clients in client-number imbalanced scenarios.
>
> In Table 1, the scenario reflects a setting where resource-rich clients are few, while resource-limited clients are the majority. Due to their small number, resource-rich clients have fewer training samples and lower participation probability (under uniform client sampling). As a result, the overall training process is dominated by resource-limited clients. This causes their average performance to appear higher, as shown in Table 1.
>
> - Performance in balanced resource distribution.
>
> To further support our hypothesis, we report results (shown in the table below) under the balanced resource setting in numbers {1,1/4,1/16,1/64}_{25,25,25,25}, following the same presentation style as Table 1, with all other experimental configurations remaining unchanged. In this case, resource-rich clients contribute more due to more frequent participation and larger models. Consequently, their local and global test accuracy is typically higher than that of lower-resource clients. This supports our understanding that the observed phenomenon in Table 1 arises from client number imbalance.
>
> - Some remarks.
>
> It is worth noting that this does not necessarily mean that the model with resource level 1 achieves the highest performance in the balanced resource setting in numbers. As presented by the related study about pruning, moderately pruned models usually can outperform unpruned ones. In our results, clients with resource level 1/4 often perform better than that of others. This observation suggests that appropriate sparsity may lead to better generalization, which is also one of the reasons why dropout has been widely adopted in practice.
>
> - Not likely caused by overfitting.
>
> We also considered whether the observed pattern could be caused by overfitting of large models. However, both local and global test accuracy curves across different resource levels show a consistent upward trend and plateauing behavior, rather than the degradation typically associated with overfitting. This suggests that overfitting is unlikely to be the main cause. Due to rebuttal space constraints, we omit the local and global test accuracy plots.
>
> We hope this explanation clarifies the underlying reasons behind the performance trend observed in Table 1.
>
> | Resource level | 1 (AccL/AccG)   | 1/4 (AccL/AccG) | 1/16 (AccL/AccG) | 1/64 (AccL/AccG) |
> | -------------- | --------------- | --------------- | ---------------- | ---------------- |
> | HeteroFL       | 32.86/31.47     | 32.68/30.89     | 26.83/26.10      | 25.12/22.98      |
> | FedRolex       | 37.11/34.47     | 36.09/32.54     | 28.17/27.82      | 20.35/17.66      |
> | ScaleFL        | 35.61/34.00     | 39.12/35.99     | 37.13/36.13      | 40.15/34.98      |
> | FIARSE         | 36.16/34.48     | 38.54/36.00     | 35.22/34.03      | 34.69/29.23      |
> | **FedLASE**    | **42.81/40.19** | **44.19/42.16** | **39.35/38.82**  | **39.24/33.94**  |
>
>
> **Response to [Q2]: Practical considerations on GPU systems.**
>
> We appreciate the reviewer’s insightful question. Compared with **unstructured submodel extraction methods** such as FIARSE, our method theoretically offers computational advantages. For **structured submodel extraction methods** like FedRolex and HeteroFL, under the same number of training rounds, they may achieve better memory usage and computation speed in practice on GPU systems due to better alignment with parallel hardware architectures. However, it is well-known that structured submodel extraction methods often suffer from lower model accuracy compared to unstructured approaches. Therefore, in real-world applications, the choice between structured and unstructured submodel extraction methods should be made based on the specific trade-offs between computational efficiency and model performance. In addition, studying unstructured pruning methods helps us gain a deeper understanding of the network training process, even if such methods may not offer advantages in terms of hardware execution.
>
> **Response to [Q3]: The significance of the new contribution.**
>
> We thank the reviewer for their comments. Below, we clarify our key contributions and explain why our method addresses challenges not considered in [21].
>
> 1. More realistic scenario considered.
>
> Our work targets a more practical federated learning scenario where resource-constrained clients are the majority, while resource-rich clients are few. In contrast, [21] focuses on balanced resource levels across clients, which is less reflective of real-world situations. This new setup introduces important challenges for collaborative training.
>
> 2. Limitations of [21] in practical settings.
>
> The FIARSE method in [21] selects parameters based on global importance ranking and allows each client to choose its own important elements. However, in our realistic setting:
>
> - Submodel structure is easily broken: Resource-limited clients can only select a small number of parameters, and global ranking often ignores structural consistency across layers. This leads to fragmented submodels and degraded local training.
>
> - Client collaboration becomes difficult: Resource-rich clients have limited training data and receive models with inconsistent structures. As a result, it becomes hard for them to benefit from collaboration with low-resource clients. This worsens client drift and slows down convergence.
>
> 3.  Simple yet effective solution.
>
> To address the above issues, we propose a layer-wise submodel extraction strategy. Instead of selecting parameters purely by importance, we control the selection within each layer to maintain model structure, which makes it (i) help preserve model architecture consistency, (ii) reduce divergence among clients, (iii) allow resource-rich clients to better learn from others, (iv) improve overall training performance in unbalanced scenarios.
>
> 4. Key insight and contribution.
>
> Our main insight is that structural inconsistency in submodels is a major cause of training divergence in realistic federated learning. A purely greedy, importance-aware approach, like in [21], can be harmful when client resources are highly unbalanced. By considering both parameter importance and structural consistency, our method achieves better collaboration among clients. This is a necessary design consideration for future FL methods aimed at realistic, heterogeneous deployments.

---

> > ### Comment · Reviewer_EUdt · 2025-08-04
> >
> > I appreciate the authors' thoughtful responses and the additional experimental results. Most of my concerns have been addressed. I will carefully consider other reviewers' comments.

---

> > > ### Author Response · Authors · 2025-08-05
> > >
> > > Dear Reviewer EUdt,
> > >
> > > Thanks for your thoughtful feedback and for considering the revisions we made. We are glad to hear that most of your concerns have been addressed. We truly appreciate your time and effort in reviewing our work. We look forward to hearing any further insights you may have, and we are happy to respond to any additional comments you might provide.
> > >
> > > Thank you once again for your valuable contribution to improving our manuscript.
> > >
> > >
> > > Best regards,
> > >
> > > Authors

---

### Official Review · Reviewer_eUKU · 2025-07-01

**Clarity:** 3
**Significance:** 2
**Originality:** 2
**Rating:** 3
**Confidence:** 4

**Summary:**

This paper focuses on the system heterogeneity across clients in Federated Learning. The authors developed a sub-model extraction method with both layer-wise and parameter-wise important, so that the extracted model can fit the resource levels on individual clients. A proof is given on the convergence of the method. Experiments are conducted on CIFAR-10 and CIFAR-100 and compared to four existing baselines.

**Questions:**

* Please clarify the difference from [21] and why the new contribution is significant
* Why the other methods for solving system heterogeneity in FL, especially [22], [23], [24], [13], are not included in the experiments?
* Would the same method work for Transformer architecture?

**Ethical Concerns:**

["NO or VERY MINOR ethics concerns only"]

**Final Justification:**

I would raise my score to weak reject to credit the theoretical contribution.
However, I still believe the real technical novelty is incremental based on the prior work, and more comprehensive experiments can be conducted.

**Limitations:**

would be good to discuss the limitation in the main paper, rather than in Appendix

**Quality:**

3

**Strengths And Weaknesses:**

Strength:
* The theoretical analysis considering the impact of system heterogeneity on FL convergence is interesting
* Experiments are done across various system and data heterogeneity

Weakness:
* The main technical contribution, i.e., considering both layer-wise and parameter-wise important during submodel extraction, is merely an incremental contribution in addition to [21]. Basically, [21] proposed a submodel extraction method based on the importance of each parameter; in this paper, the authors extended this method to consider layer importance
* The experiments on CIFAR-10 and CIFAR-100 using ResNets is very limited in terms of the diversity of the applications and network structures. The performance under larger scale datasets and Transformer architectures are unknown
* As the authors mentioned, [22],  [23], [13], [24] are all designed for solving the same system heterogeneity issue in FL. However, these methods are not compared against during the experiments, which makes the empirical results weaker.

---

> ### Author Rebuttal · Authors · 2025-07-31
>
> **Response to [Q1] and [W1]: The significance of the new contribution.**
>
> We thank the reviewer for their comments. Below, we clarify our key contributions and explain why our method addresses challenges not considered in [21].
>
> 1. More realistic scenario considered.
>
> Our work targets a more practical federated learning scenario where resource-constrained clients are the majority, while resource-rich clients are few. In contrast, [21] focuses on balanced resource levels across clients, which is less reflective of real-world situations. This new setup introduces important challenges for collaborative training.
>
> 2. Limitations of [21] in practical settings.
>
> The FIARSE method in [21] selects parameters based on global importance ranking and allows each client to choose its own important elements. However, in our realistic setting:
>
> - Submodel structure is easily broken: Resource-limited clients can only select a small number of parameters, and global ranking often ignores structural consistency across layers. This leads to fragmented submodels and degraded local training.
>
> - Client collaboration becomes difficult: Resource-rich clients have limited training data and receive models with inconsistent structures. As a result, it becomes hard for them to benefit from collaboration with low-resource clients. This worsens client drift and slows down convergence.
>
> 3.  Simple yet effective solution.
>
> To address the above issues, we propose a layer-wise submodel extraction strategy. Instead of selecting parameters purely by importance, we control the selection within each layer to maintain model structure, which makes it (i) help preserve model architecture consistency, (ii) reduce divergence among clients, (iii) allow resource-rich clients to better learn from others, (iv) improve overall training performance in unbalanced scenarios.
>
> 4. Key insight and contribution.
>
> Our main insight is that structural inconsistency in submodels is a major cause of training divergence in realistic federated learning. A purely greedy, importance-aware approach, like in [21], can be harmful when client resources are highly unbalanced. By considering both parameter importance and structural consistency, our method achieves better collaboration among clients. This is a necessary design consideration for future FL methods aimed at realistic, heterogeneous deployments.
>
> **Response to [Q2] and [W3]: Comparison with [22], [23], [13], [24].**
>
> We appreciate the reviewer’s comment regarding comparisons with related works addressing system heterogeneity. We provide detailed clarifications for each of these methods below:
>
> - Depth [13]: Depth extracts submodels purely based on network depth. However, in realistic cross-device scenarios with a large proportion of resource-constrained clients (e.g., minimum resource level 1/64), depth-based truncation fails to generate valid submodels under such tight constraints. For instance, truncating ResNet-18 by depth alone cannot produce functional submodels small enough to fit these devices while retaining meaningful structure. In contrast, our method is more flexible and adapts submodel extraction to diverse resource profiles, making it better suited for real-world settings dominated by resource-limited clients.
>
> - FjORD [24]: FIARSE has already demonstrated superior performance over FjORD [24], as reported in [21]. Building on FIARSE, our method further achieves significant improvements across diverse heterogeneous scenarios, demonstrating clear advancements beyond [21].
> Additionally, FjORD did not release official code, and its implementation details are insufficiently specified, making faithful reproduction challenging and potentially unreliable. Therefore, we focus on FIARSE—an importance-aware submodel extraction baseline with publicly available code—as the most relevant and fair baseline for empirical comparison.
>
> - FedLMT [23]: FedLMT uses low-rank decomposition, but its applicable resource range is constrained by architectural limits. For example, with ResNet-18, the lowest feasible rank ratio is $\alpha \geq 0.015625$. As noted in [23], for a resource level of 1/4, FedLMT requires $\alpha = 0.03$; consequently, for 1/16 resources, $\alpha$ must be less than 0.0075, which contradicts the condition $\alpha \geq 0.015625$. This limitation prevents FedLMT from accommodating extreme yet realistic heterogeneous scenarios, such as when devices differ widely in computational capacity (e.g., 1 vs. 1/64).
> Moreover, enforcing a rank ratio at the limit severely constrains model capacity, undermining its training effectiveness. Due to these constraints, FedLMT is unsuitable for the highly resource-imbalanced scenarios targeted in our work, and direct comparison would be neither practical nor meaningful.
>
> - FedDSE [22]: Since the code for this work was not publicly available, we did not include it in our initial comparisons. Recently, we obtained the code from the authors and conducted experiments under the data heterogeneity setting L=4 as considered in their paper. The results shown below illustrate that our method significantly outperforms FedDSE across various system heterogeneity scenarios, demonstrating the superiority of our layer-adaptive submodel extraction approach. Moreover, our method can also be combined with FedDSE to design more effective layer pruning ratios for FedDSE.
>
>
> - Justification of our experimental focus on FIARSE[21]: Given these considerations, we focus on FIARSE, which (i) is reproducible, (ii) is recognized as a strong and recent baseline for importance-aware submodel extraction, and (iii) directly aligns with our method’s design goals. Importantly, our method consistently outperforms FIARSE across various heterogeneous settings, demonstrating substantial empirical gains. This provides strong evidence of our contribution even without direct comparisons to methods that are either infeasible or not reproducible in our targeted scenarios.
>
> We will explicitly add these clarifications in the revised manuscript and, where feasible, provide supplementary experiments or discussion to better position our method relative to these works.
>
> | Scenario    |  {10,20,30,40} | {25,25,25,25} |
> | ----------- | ------------- | ------------- |
> | FedDSE      | 6.72            | 7.98             |
> | **FedLASE** |**19.62**            | **22.15**          |
>
> **Response to [Q3] and [W2]: Additional experiments on NLP tasks.**
>
> Thank you for your insightful comment regarding the generalizability of our proposed method. We fully agree that evaluating on more diverse tasks and model architectures is crucial for demonstrating broader applicability. To address this concern, we extend our experiments to the AGNews dataset, a widely used large-scale text classification benchmark in the NLP domain. We adopt a pre-trained RoBERTa-based transformer model as the backbone. The training is conducted with $2$ local epochs, $300$ global rounds, a learning rate of $0.0001$, using the AdamW optimizer with momentum parameters of $(0.9, 0.95)$, and a batch size of $20$. We compare FedLASE with FIARSE, which is the most competitive baseline. As shown in the following table, FedLASE consistently outperforms FIARSE in this NLP setting as well, achieving both higher global accuracy and better client-level fairness. These results further validate the generalizability and effectiveness of FedLASE beyond small-scale image classification and convolutional architectures.
> | Scenario    | {5,10,25,60}, IID | {5,10,25,60}, Dir(1) | {10,20,30,40}, IID |{10,20,30,40}, Dir(1) |
> | ----------- | ----------------- | -------------------- | ------------------ | -------------------- |
> | FIARSE      | 30.54             | 38.47                | 59.80              | 55.67                |
> | **FedLASE** | **64.10**          | **70.93**            | **68.21**          | **70.96**            |
>
> Notes: All experiments are conducted for CIFAR-100 under ResNet-18 using the following default settings: learning rate is $0.01$, momentum is $0.8$, the training round is $T=500$. Other settings remain the same as described in the main paper.
>
> **Response to [Limitations]**
>
> Thank you for the valuable suggestion. We agree that discussing the limitations in the main paper will provide clearer insights and benefit readers’ understanding. We will revise the manuscript to move the discussion of limitations from the appendix to the main text in the next version.

---

> ### Comment · Reviewer_eUKU · 2025-08-03
> **Response**
>
> Thanks the authors for the response and the additional experiments results. The efforts are highly appreciated.
>
> However, my following concerns remain:
> 1. Significance of the contribution compared to [21].
> While the authors provide a thorough comparison against [21], the fact remains that the main proposal is an extension of [21]. Essentially, this paper focuses on a narrower niche of system heterogeneity—specifically, more “skewed” resources, with a larger proportion of constrained, small devices. In practical deployments, it is not entirely clear how common such a significantly “skewed” environment is.
>
> In addition, prior work has studied FL under skewed system heterogeneity [1]. Why not simply using these techniques?
>
> [1] Zhang, Jiayun, et al. "How few davids improve one goliath: Federated learning in resource-skewed edge computing environments." Proceedings of the ACM Web Conference 2024. 2024.
>
> 2. Lack of comparison.
> While the authors provide more discussions and results, the paper would be stronger and more convincing with a comprehensive comparison.

---

> > ### Author Response · Authors · 2025-08-09
> >
> > We thank the reviewer for the constructive feedback. Nevertheless, we have evaluated FedLASE under a wide range of system heterogeneity scenarios and demonstrated its superiority over the most closely related methods, which we believe sufficiently supports the core message of our work.

---

### Official Review · Reviewer_iCC5 · 2025-07-02

**Clarity:** 3
**Significance:** 2
**Originality:** 2
**Rating:** 3
**Confidence:** 3

**Summary:**

Federated Learning (FL) faces challenges due to system heterogeneity among edge devices, leading to performance discrepancies in existing submodel extraction methods. To address this, the authors propose ​​FedLASE​​, a ​​Layer-Adaptive Submodel Extraction​​ framework that intelligently selects critical parameters while maintaining model structural integrity. FedLASE quantifies layer importance, hierarchically extracts key parameters under resource constraints, and ensures balanced performance across heterogeneous clients. Theoretical analysis confirms its convergence and robustness under system heterogeneity. Experiments show FedLASE outperforms state-of-the-art methods in diverse scenarios.

**Questions:**

Please refer to the weakness part.

**Ethical Concerns:**

["NO or VERY MINOR ethics concerns only"]

**Final Justification:**

The authors have addressed most of my concerns. I still believe that the experimental section would benefit from evaluating FedLASE with more diverse datasets and model architectures.

**Limitations:**

Please refer to the weakness part.

**Quality:**

3

**Strengths And Weaknesses:**

Strengths:
1. This paper introduces ​​FedLASE​​, a ​​layer-adaptive submodel extraction​​ method for FL, which is an interesting idea.
2. This paper provides ​​convergence guarantees​​ and analyzes system heterogeneity impact.
3. Extensive experiments demonstrate the superiority of FedLASE.

Weaknesses:
1. In line 57, the authors mention that "this method overlooks inter-layer discrepancies." Could they provide more details on this observation? Additionally, why are inter-layer discrepancies important? This statement currently lacks quantitative evaluation.

2. In line 146, the paper defines layer importance as the mean parameter magnitude per layer, normalized via log transformation. Could the authors elaborate on why they chose to define the importance score in this way? A discussion of the motivation behind this equation would be helpful.

3. The paper's evaluation is limited to small-scale image classification tasks (CIFAR-10/100) using only convolutional architectures (ResNet-18), which may not fully demonstrate the method's generalizability to more complex and practical scenarios. There is no validation on larger-scale datasets, other architectures such as Vision Transformers (ViTs), or NLP tasks.

---

> ### Author Rebuttal · Authors · 2025-07-31
>
> **Response to [Q1]: Clarification and quantitative evidence for "inter-layer discrepancies".**
>
> (1) Empirical observation of inter-layer discrepancies
>
> FIARSE measures parameter importance based solely on parameter magnitude. To analyze layer-wise differences, we computed the maximum and mean of parameter magnitudes within each layer (reported in the first two rows of the table below, excluding the first and last layers of ResNet-18, normalization layers, and biases). As shown, the maximum parameter magnitudes differ significantly across layers, and several layers exhibit mean magnitudes that diverge substantially from others. Additionally, the number of parameters varies widely across layers. Ranking all parameters globally, as FIARSE does, overlooks both (i) inter-layer differences in parameter distributions and (ii) disparities in parameter counts. This can potentially remove critical information from important layers or disrupt the overall structural balance of the network.
>
> (2) Training dynamics: layer-wise extraction and parameter updates
>
> We further compared FIARSE and our proposed FedLASE in terms of layer extraction ratio dynamics and cumulative parameter update counts during training (reported in the last two rows of the table), and observed the following:
>
> - FIARSE maintains largely stable extraction ratios across rounds, which likely leads to repeated updates of the same subset of parameters while many others remain insufficiently trained.
>
> - Our method FedLASE, by incorporating layer structural information into submodel extraction, adaptively adjusts extraction ratios. The higher variance in extraction ratios across rounds allows more parameters to participate in training, resulting in a more balanced update distribution. This is further confirmed by our cumulative parameter update count analysis (visual comparisons omitted here due to rebuttal limitations).
>
> (3) Performance validation
>
> The superior performance of FedLASE over FIARSE further demonstrates the importance of accounting for inter-layer discrepancies. By dynamically balancing parameter training across layers, FedLASE achieves improved convergence and accuracy.
>
> | Layer $l_i$                          | 1            | 2            | 3            | 4            | 5            | 6            | 7            | 8            | 9            |
> | ------------------------------------ | ------------ | ------------ | ------------ | ------------ | ------------ | ------------ | ------------ | ------------ | ------------ |
> | $\max_j\|\theta^t_{l_i,j} \|$        | 0.33         | 0.19         | 0.19         | 0.17         | 0.15         | 0.13         | **0.35**     | 0.13         | 0.11         |
> | $mean_j\|\theta^t_{l_i,j} \|$ | 0.03         | 0.03         | 0.03         | 0.03         | 0.03         | 0.02         | **0.07**     | 0.02         | 0.02         |
> | ratio_std for FIARSE                 | 0.002854     | 0.002433     | 0.001584     | 0.001970     | 0.001215     | 0.000516     | 0.019432     | 0.000519     | 0.000410     |
> | **ratio_std for FedLASE**            | **0.009770** | **0.008375** | **0.003083** | **0.004122** | **0.003386** | **0.001963** | **0.022406** | **0.000662** | **0.001004** |
>
> | Layer $l_i$                          | 10           | 11           | 12           | 13           | 14           | 15           | 16           | 17           | 18           | 19           |
> | ------------------------------------ | ------------ | ------------ | ------------ | ------------ | ------------ | ------------ | ------------ | ------------ | ------------ | ------------ |
> | $\max_j\|\theta^t_{l_i,j} \|$        | 0.11         | 0.10         | 0.19         | 0.08         | 0.08         | 0.08         | **0.07**     | 0.16         | 0.08         | **0.06**     |
> | $mean_j\|\theta^t_{l_i,j} \|$ |  0.02        | 0.01         | **0.05**     | 0.01         | 0.01         | 0.01         | **0.01**     | 0.04         | 0.01         | 0.01         |
> | ratio_std for FIARSE                 | 0.000559     | 0.000306     | 0.016768     | 0.000380     | 0.000476     | 0.000258     | 0            | 0.009479     | 0.000494     | 0.000291     |
> | **ratio_std for FedLASE**            | **0.000937** | **0.000448** | **0.009027** | **0.000250** | **0.000264** | **0.000405** | **0.000589** | **0.005251** | **0.000513** | **0.000598** |
>
> The experiments in the first two rows of this table are conducted under the system {1}_{100}. mean_j|\theta^t_{l_i,j} | and max_j|\theta^t_{l_i,j}| with t=599 represent the maximum and mean of parameter magnitudes within $l_i$-th layer in round t, respectively. 'ratio_std' denotes the standard deviation of submodel extraction ratios over the last 600 training rounds.
>
> Notes: Unless otherwise specified, all experiments are conducted for CIFAR-100 under ResNet-18 using the following default settings: heterogeneous system is {1, 1/4,1/16,1/64}_{10,20,30,40}, learning rate is $0.01$, momentum is $0.8$, the training round is $T=1000$, and the data distribution is Dir(0.3). Other settings remain the same as described in the main paper.
>
> **Response to [Q2]: Design for the layer importance score.**
>
> - **The use of mean:** How to effectively aggregate parameter importance into a meaningful layer-wise importance score is a critical design choice. A few straightforward strategies include **computing the mean (FedLASE), maximum (FedLASE w max), or assigning equal extraction ratios (FedLASE w eq)** to all layers. Through extensive experiments, we found that using the **average importance** for the parameters within that layer provides a more stable and representative measure of that layer’s contribution. Some representative experimental results are shown in the following table.
>
> - **The use of logarithmic transformation:** Once the preliminary layer importance scores are computed, a new challenge may be encountered: if a layer has a substantially larger number of parameters and correspondingly higher layer importance, it may dominate the submodel extraction process. As a result, layers with relatively lower total importance may end up with very few preserved parameters, potentially leading to structural imbalance and damage to the overall network architecture. To mitigate this issue, we apply a **logarithmic transformation** to smooth the importance scores across layers. This helps reduce the variance and prevents drastic parameter imbalance. The ablation results presented in the following table further demonstrate the effectiveness of this smoothing strategy and highlight the critical role of properly normalized layer importance scores.
>
> - **Other functions:** Although **linear normalization** might seem simpler, it often lacks the desired smoothing effect, and its hyperparameter tuning can be non-trivial. Alternatively, **learning a non-linear mapping function** from data is a possible solution, but raises further challenges in terms of privacy leakage, dataset selection, and computational cost. Based on these observations, we adopt a log-based heuristic function, which is both simple and effective, to define layer importance. Designing more principled and adaptive layer scoring functions using richer information remains an important direction for future work.
>
> | Scenario        | {5,10,25,60} | {10,20,30,40} | {25,25,25,25} |
> | --------------- | ------------ | ------------- | ------------- |
> | FedLASE w eq    | 35.32        | 38.47         | 38.79         |
> | FedLASE w max   | 29.73        | 34.70         | 38.14         |
> | FedLASE w/o log | 36.98        | 37.96         | 39.09         |
> | **FedLASE (w mean & log)**     | **40.22**    | **41.25**     | **42.73**     |
>
> **Response to [Q3]: Additional experiments on NLP tasks.**
>
> Thank you for your insightful comment regarding the generalizability of our proposed method. We fully agree that evaluating on more diverse tasks and model architectures is crucial for demonstrating broader applicability. To address this concern, we extend our experiments to the AGNews dataset, a widely used large-scale text classification benchmark in the NLP domain. We adopt a pre-trained RoBERTa-based transformer model as the backbone. The training is conducted with $2$ local epochs, $300$ global rounds, a learning rate of $0.0001$, using the AdamW optimizer with momentum parameters of $(0.9, 0.95)$, and a batch size of $20$. We compare FedLASE with FIARSE, which is the most competitive baseline. As shown in the following table, FedLASE consistently outperforms FIARSE in this NLP setting as well, achieving both higher global accuracy and better client-level fairness. These results further validate the generalizability and effectiveness of FedLASE beyond small-scale image classification and convolutional architectures.
> | Scenario    | {5,10,25,60}, IID | {5,10,25,60}, Dir(1) | {10,20,30,40}, IID |{10,20,30,40}, Dir(1) |
> | ----------- | ----------------- | -------------------- | ------------------ | -------------------- |
> | FIARSE      | 30.54             | 38.47                | 59.80              | 55.67                |
> | **FedLASE** | **64.10**          | **70.93**            | **68.21**          | **70.96**            |

---

> > ### Comment · Reviewer_iCC5 · 2025-08-06
> >
> > Thank you for your responses. I still believe that the experimental section would benefit from evaluating FedLASE with more diverse datasets and model architectures. I am inclined to keep my original score.

---

> > > ### Author Response · Authors · 2025-08-09
> > >
> > > We sincerely thank the reviewer for the thoughtful feedback and for taking the time to assess our work. In the future work, we plan to extend our experiments to additional datasets and larger-scale architectures to further validate the generality of FedLASE. We appreciate your perspective, which we believe will help improve the comprehensiveness of our evaluation.

---

> ### Author Response · Authors · 2025-08-05
> **We look forward to your acknowledgement**
>
> Dear Reviewer iCC5:
>
> We hope this message finds you well. We would like to express our sincere gratitude for your insightful and constructive comments on our manuscript. As we approach the conclusion of the author-reviewer discussion period, we kindly request that you review our response and let us know if it adequately addresses your concerns. If you require any further information or clarification, please feel free to contact us. We would be glad to provide any additional details you may need.
>
> Thank you once again for your time and valuable feedback.
>
> Best regards,
> Authors

---

### Official Review · Reviewer_1hNE · 2025-07-04

**Clarity:** 2
**Significance:** 2
**Originality:** 2
**Rating:** 4
**Confidence:** 3

**Summary:**

The paper proposes FedLASE, a method for system-heterogeneous FL that extracts submodels for clients based on layer-wise importance and parameter magnitude. The key idea is to preserve critical structure across clients with different resource capacities, aiming to avoid the imbalance often seen when smaller models dominate the optimization due to being deployed on more numerous low-resource devices.

**Questions:**

* What is the evidence (empirical or theoretical) that post-aggregation magnitude is a reliable proxy for parameter importance in non-IID FL settings?

* Why was logarithmic layer normalization chosen over alternatives? Was this choice validated experimentally?

* How much of the performance gain can be attributed to the layer-wise extraction logic, versus STE, versus full-layer retention, versus personalized aggregation? Can you provide a detailed ablation study to evaluate the different proposed methods?

* Can you report confidence intervals or standard deviations for Table 1 and Table 2 to justify significance?

* How would FedLASE handle online client dropouts or dynamic system profiles, a core part of real-world FL?

**Ethical Concerns:**

["NO or VERY MINOR ethics concerns only"]

**Limitations:**

Partially addressed. While the paper includes a limitations section, it downplays the core reliance on heuristics and avoids deeper discussion of the design tradeoffs and scalability bottlenecks. A more honest reflection on the limitations of FedLASE's static parameter ranking in dynamic FL environments would be welcome.

**Paper Formatting Concerns:**

No concern.

**Quality:**

3

**Strengths And Weaknesses:**

Strengths:

+ The performance gap caused by system heterogeneity is real and often overlooked. The problem definition is relevant and impactful.

+ The use of normalized per-layer importance to guide submodel construction is a simple yet effective design that improves upon one-size-fits-all global ranking.

+ The convergence analysis under client-specific model reduction noise is non-trivial and appears correct.


Weaknesses:

- The paper heavily relies on heuristics without clear justification. There is no ablation or theoretical rationale for these design choices. Why logarithmic normalization? Why not simpler linear scaling? These design decisions appear ad hoc.

- You assume that parameter magnitude after aggregation correlates with utility, which may not hold under non-IID data and heterogeneous participation. This assumption is not theoretically or empirically verified. If the importance is stale or misaligned, the resulting submodels could be suboptimal.

- The method bundles together several mechanisms, layer-aware pruning, full retention of specific layers, STE tricks, personalized aggregation, yet the paper does not isolate which of these components drive the observed gains. Without proper ablation, it is unclear how much is due to FedLASE’s core idea.

- Despite claiming significant and consistent improvements, no error bars or confidence intervals are reported. Given the random client selection and data heterogeneity, this is a major omission that undermines the credibility of the empirical claims.

- The experiments are limited to 100 clients and synthetic partitions. It is unclear whether FedLASE generalizes to real-world cross-device settings with more clients and online availability constraints.

---

> ### Author Rebuttal · Authors · 2025-07-31
>
> **Response to [Q1] and [W2]: Proxy for parameter importance.**
>
> We sincerely thank the reviewer for raising all these insightful and interesting questions. Using the parameter magnitude as a proxy for parameter importance has been widely adopted in the area of pruning for both centralized and federated learning, due to its simplicity and practical effectiveness.
>
> - **Theoretically,** this choice can be motivated through a second-order Taylor expansion of the global loss function in federated learning:
>
> $F(\theta + \Delta\theta) \approx F(\theta) + \nabla F(\theta)^T \Delta\theta + \frac{1}{2} \Delta\theta^T H \Delta\theta$
>
> with $H$ being the corresponding Hessian matrix. In typical federated optimization settings, local models are assumed to converge reasonably well before aggregation, implying $\nabla F(\theta) \approx 0$. Then, the loss difference is dominated by $\Delta\theta^T H \Delta\theta$. If we further assume $H \approx I$, then the impact of zeroing the $i$-th parameter $\theta_i$ can be approximated as $\theta_i^2$, suggesting that parameters with smaller magnitude usually have less contribution to the loss and can be safely pruned.
>
> - **Empirically,** we adopt the same strategy as FIARSE [21], which also uses post-aggregation parameter magnitude to guide submodel construction under heterogeneity. Their results, along with our experiments, demonstrate that this simple metric is robust across many settings, even though it may not be exact.
>
> **Response to [Q2] and [W1]: Reason for selecting logarithmic layer normalization.**
>
> - **Reason:** After the predefined layer importance scores are computed, a new challenge may be encountered: if a layer has a substantially larger number of parameters and correspondingly higher layer importance, it may dominate the submodel extraction process. As a result, layers with relatively lower total importance may end up with very few preserved parameters, potentially leading to structural imbalance and damage to the overall network architecture. To mitigate this issue, we apply a **logarithmic transformation** to smooth the importance scores across layers. This helps reduce the variance and prevents drastic parameter imbalance.
>
> - **Ablation study:** The ablation results shown in the following response further demonstrate the effectiveness of this smoothing strategy and highlight the critical role of properly normalized layer importance scores.
>
> - **Other selection:** Although **linear normalization** might seem simpler, it often lacks the desired smoothing effect, and its hyperparameter tuning can be non-trivial. Alternatively, **learning a non-linear mapping function** from data is a possible solution, but raises further challenges in terms of privacy leakage, dataset selection, and computational cost. Based on these observations, we adopt a log-based function, which is both simple and effective. Designing more principled and adaptive layer scoring functions using richer information remains an important direction for future work.
>
> **Response to [Q3] and [W3]: Ablation study.**
>
> To better understand the contributions of different components in FedLASE, we conducted additional experiments that isolate the effects of: (i) STE-based gradient approximation, (ii) personalized overlapping aggregation, (iii) full retention of specific layers, and (iv) logarithmic layer normalization. Specifically, we evaluate the following variants:
> - FedLASE w/o STE: removes the STE trick used during submodel training.
> - FedLASE w/o overlap: replaces the personalized aggregation (which uses overlapping weighted averaging of masked parameters) with a FedAvg-style update: $\theta = \frac{1}{|\mathcal{A}|} \sum_{n \in \mathcal{A}} M^n \odot \theta^n$.
> - FedLASE w all layers: extracts submodel extraction across all layers (excluding batch normalization and bias terms), instead of selectively preserving key layers.
> - FedLASE w/o log: disables the logarithmic layer normalization when computing layer importance.
>
> Notably, for fair comparison with FIARSE [21], we retain STE and overlapping aggregation in our default setup. This allows us to isolate the impact of our proposed layer-aware extraction scheme. We can see from the table below that:
> - Removing logarithmic normalization will bring a large performance drop, yet 'FedLASE w/o log' still significantly outperforms FIARSE, demonstrating the importance of processing layer-level statistics beyond raw parameter magnitudes.
> - 'FedLASE w all' shows slightly lower performance than our default version, suggesting that retaining the first and last layers intact indeed helps preserve critical information necessary for stable training and final prediction. Of course, we can also fix the extraction ratios of the first and last layers. However, to avoid introducing additional hyperparameters, we retain all parameters in these layers.
> - The other two variants (w/o STE, w/o overlap) exhibit similar trends to those for FIARSE, reaffirming that our improvements stem from our core contribution, i.e., the sample and effective layer-adaptive submodel extraction framework for heterogeneous FL.
>
> | Scenario            | {5,10,25,60} | {10,20,30,40} | {25,25,25,25} |
> | ------------------- | ------------ | ------------- | ------------- |
> | FIARSE w/o overlap  | 32.55        | 34.94         | 36.96         |
> | FIARSE w/o STE      | 29.49        | 33.50         | 37.15         |
> | **FIARSE**          | **28.65**    | **33.50**     | **36.69**     |
> | FedLASE w/o overlap | 40.91        | 41.87         | 42.84         |
> | FedLASE w/o STE     | 39.82        | 42.27         | 41.73         |
> | FedLASE w all layers| 39.41        | 39.86         | 39.89         |
> | FedLASE w/o log     | 36.98        | 37.96         | 39.09         |
> | **FedLASE**         | **40.22**    | **41.25**     | **42.73**     |
>
> Notes: Unless otherwise specified, all experiments are conducted under the following default settings: heterogeneous system is {1, 1/4,1/16,1/64}_{10,20,30,40}, learning rate is $0.01$, momentum is $0.8$, the training round is $T=1000$, and the data distribution is Dir(0.3). Other settings remain the same as described in the main paper.
>
> **Response to [Q4] and [W4]: Impact of random.**
>
> Thank you for pointing this out. All results in Tables 1 and 2 are averaged over three different random seeds. We agree that it is important to understand the robustness of the proposed method under different sources of randomness, such as client sampling, data partitioning, and initialization. Due to space constraints in the rebuttal, we evaluate FedLASE and the baselines under more independent random seeds to more clearly demonstrate the impact of randomness. For each method, we report the global test accuracy under each seed, along with the corresponding mean and standard deviation. The results show that while there is some variance due to randomness, our method consistently outperforms FIARSE and other baselines across all seeds. This confirms the stability and robustness of FedLASE under realistic federated conditions.
> | Seed        | 0         | 1         | 42        | 100       | 2025      | 6666      | mean      | std      |
> | ----------- | --------- | --------- | --------- | --------- | --------- | --------- | --------- | -------- |
> | HeteroFL    | 26.79     | 28.43     | 28.63     | 28.40     | 27.69     | 27.35     | 27.88     | 0.73     |
> | FedRolex    | 29.11     | 28.98     | 28.81     | 28.87     | 28.74     | 26.86     | 28.56     | 0.84     |
> | ScaleFL     | 33.39     | 33.79     | 33.17     | 32.35     | 32.77     | 31.36     | 32.81     | 0.87     |
> | FIARSE      | 33.50     | 33.60     | 35.25     | 34.38     | 35.12     | 33.04     | 34.15     | 0.91     |
> | **FedLASE** | **41.25** | **41.68** | **42.25** | **42.32** | **41.37** | **42.27** | **41.86** | **0.49** |
>
> **Response to [Q5] and [W5]: Scenarios for online client dropouts.**
>
> - **Results in Tables 1 and 2:** We thank the reviewer for raising the important issue of online client availability, which is indeed a key challenge in real-world federated learning systems. In our original submission, we have already included experiments simulating typical client dropout scenarios: specifically, we considered 100 clients, with only 10% randomly selected in each round to simulate partial availability. These results are shown in Tables 1 and 2 and demonstrate the effectiveness of FedLASE.
> To further explore more realistic and complex system dynamics, we conduct two additional experiments to reflect heterogeneous and time-varying participation:
> - **Static scenario:** We increase the total number of clients to 200 and randomly select 10% (i.e., 20 clients) to participate in each round.
> - **Dynamic scenario:** We again use 200 clients, but the client participation ratio is dynamically sampled from {10%, 15%, 20%} in each round, simulating more complex real-world deployment scenarios with fluctuating client participation.
> The results in the following table illustrate that our method still consistently outperforms all baselines across both static and dynamic participation settings, highlighting its robustness to client dropout and its adaptability to more practical FL environments.
>
> | Scenario    | Static    | Dynamic   |
> | ----------- | --------- | --------- |
> | HeteroFL    | 27.16     | 26.71     |
> | FedRolex    | 26.78     | 27.20     |
> | ScaleFL     | 31.82     | 31.34     |
> | FIARSE      | 31.96     | 31.25     |
> | **FedLASE** | **37.19** | **37.12** |
>
> **Response to [Limitations]**
>
> Although our proposed strategy is simple and effective, designing better alternatives to the logarithmic layer normalization function remains a challenging and open problem. Moreover, incorporating system heterogeneity information to design more fine-grained and adaptive metrics for parameter and layer importance is a promising direction for future research. We will include these limitations in the revised manuscript.

---

> ### Author Response · Authors · 2025-08-05
>
> Dear Reviewer 1hNE:
>
> We hope this message finds you well. We would like to express our sincere gratitude for your insightful and constructive comments on our manuscript. As we approach the conclusion of the author-reviewer discussion period, we kindly request that you review our response and let us know if it adequately addresses your concerns. If you require any further information or clarification, please feel free to contact us. We would be glad to provide any additional details you may need.
>
> Thank you once again for your time and valuable feedback.
>
> Best regards,
>
> Authors

---

### Note · Authors · 2025-08-14

For the more **realistic and meaningful** system-heterogeneous scenario considered in our work **(1hNE)**, we propose a **simple yet effective (1hNE) and interesting (iCC5)** FL method, supported by **non-trivial (1hNE) / interesting (eUKU) convergence analysis**, and validated by **extensive experiments across various system and data heterogeneity (iCC5, eUKU)**.
We sincerely thank all reviewers for their valuable feedback. Below is the summary of our responses.

**1.Clarification of Novelty and Contribution (eUKU, EUdt)**

Several reviewers noted similarities to FIARSE, but our main contribution targets realistic resource-skewed FL, where low-resource clients dominate. In such cases, global ranking causes structural inconsistency and degraded collaboration. Our layer-adaptive extraction preserves the overall model architecture, reducing divergence and improving cross-client knowledge transfer, as confirmed by experiments. Non-trivial convergence analysis is also given.

**2. Parameter Importance and Layer Normalization (1hNE, iCC5)**

We justify using parameter magnitude via theoretical analysis and prior evidence. Logarithmic normalization is simple and effectively prevents large layers from dominating while preserving architectural balance; ablations confirm its effectiveness.

**3. Ablation Studies (1hNE)**

We conducted ablations by removing STE, overlapping aggregation, full-layer retention, or log-normalization. The first two are retained for consistency with FIARSE, while the latter two yield substantial performance improvements.

**4. Additional Experiments for Generalizability (iCC5, eUKU)**

We extended evaluation to the AGNews dataset with a RoBERTa backbone under various resource-heterogeneous settings. FedLASE achieved substantial gains over FIARSE, demonstrating its scalability to larger or more complex datasets and models.

**5. Broader Baseline Comparisons (eUKU)**

We clarified why certain baselines ([22,23,24,13]) were excluded initially: some baselines are infeasible under the scenarios we considered, lack reproducible code, or are already outperformed by FIARSE. We later added FedDSE [22], showing substantial gains, and focused on relevant baselines for realistic heterogeneous settings.

**6. Limitations and Future Work (1hNE, eUKU)**

FedLASE uses a heuristic scoring metric for layer extraction. Future work will explore principled, adaptive metrics with richer heterogeneity information and apply them to larger models and datasets.

---

### Decision · Program_Chairs · 2025-09-17

**Decision:**

Reject

**Comment:**

The paper proposes a new method for heterogeneous Federated Learning which leverages layer-wise importance and parameter magnitude to extract submodels for clients. The paper presents theoretical analysis for its convergence and robustness under various system heterogeneity. Experimental evaluations also show the method outperforms other state-of-the-art methods. However, multiple reviewers have raised the concerns about the technical novelty over existing works such as FIARSE, as well as the insufficient evaluations of existing baselines. Although the authors argued that the proposed method targets a more realistic setting, the actual technical contribution appears indeed a bit limited.

During the rebuttal phase, the authors provided some additional experimental results, but multiple reviewers pointed out that more comprehensive experiments are still yet to conducted. Therefore it appears that the paper at its current status doesn't meet the standard of completeness that NeurIPS requires.  It will improve greatly after another round of revision by incorporating reviewers' suggestions.